# Capturing the songs of mice with an improved detection and classification method for ultrasonic vocalizations (BootSnap)

Reyhaneh Abbasi[1,2,3]*, Peter Balazs[1], Maria Adelaide Marconi[2], Doris Nicolakis[2], Sarah M. Zala[2ᵒ], Dustin J. Penn[2ᵒ]

**1** Acoustic Research Institute, Austrian Academy of Sciences, Vienna, Austria, **2** Konrad Lorenz Institute of Ethology, Department of Interdisciplinary Life Sciences, University of Veterinary Medicine, Vienna, Austria, **3** Vienna Doctoral School of Cognition, Behaviour and Neuroscience, University of Vienna, Vienna, Austria

ᵒ These authors contributed equally to this work.

* reyhaneh.abbasi@oeaw.ac.at

**Data Availability Statement:** The sound files of wild mice that we used to evaluate our model is available online ((https://mousetube.pasteur.fr/) or

## Abstract

House mice communicate through ultrasonic vocalizations (USVs), which are above the range of human hearing (>20 kHz), and several automated methods have been developed for USV detection and classification. Here we evaluate their advantages and disadvantages in a full, systematic comparison, while also presenting a new approach. This study aims to 1) determine the most efficient USV detection tool among the existing methods, and 2) develop a classification model that is more generalizable than existing methods. In both cases, we aim to minimize the user intervention required for processing new data. We compared the performance of four detection methods in an out-of-the-box approach, pretrained DeepSqueak detector, MUPET, USVSEG, and the Automatic Mouse Ultrasound Detector (A-MUD). We also compared these methods to human visual or 'manual' classification (ground truth) after assessing its reliability. A-MUD and USVSEG outperformed the other methods in terms of true positive rates using default and adjusted settings, respectively, and A-MUD outperformed USVSEG when false detection rates were also considered. For automating the classification of USVs, we developed *BootSnap* for supervised classification, which combines bootstrapping on Gammatone Spectrograms and Convolutional Neural Networks algorithms with Snapshot ensemble learning. It successfully classified calls into 12 types, including a new class of false positives that is useful for detection refinement. *BootSnap* outperformed the pretrained and retrained state-of-the-art tool, and thus it is more generalizable. *BootSnap* is freely available for scientific use.

## Author summary

House mice and many other species use ultrasonic vocalizations to communicate in various contexts including social and sexual interactions. These vocalizations are increasingly investigated in research on animal communication and as a phenotype for studying the genetic basis of autism and speech disorders. Because manual methods for analyzing

(https://zenodo.org/record/5771669#.
YiohQ9XML3g)). The sound files of laboratory
mice, uploaded by Chabout et al. (2015), are
already available online ((https://duke.box.com/
shared/static/6j08fzyto8nuxxstk6bcpi9n52bk5bu4.
wav) and (https://duke.box.com/shared/static/
y5o7zw8jx9ugb2qocozyaup7xlby1sr8.wav)).
Codes needed to reproduce our results are
available online (https://github.com/
ReyhanehAbbasi/BootSnap).

**Funding:** This work was supported by the FWF P
34922-N project ("NoMASP: Nonsmooth
Nonconvex Optimization Methods for Acoustic
Signal Processing") to PB and by a grant (FWF P
28141-B25) of the Austrian Science Foundation
(http://www.fwf.ac.at) to DJP and SMZ. The
funders had no role in study design, data collection
and analysis, decision to publish, or preparation of
the manuscript.

**Competing interests:** The authors have declared
that no competing interests exist.

vocalizations are extremely time consuming, automatic tools for detection and classification are needed. We evaluated the performance of the available tools for analyzing ultrasonic vocalizations, and we compared detection tools for the first time to manual methods ("ground truth") using recordings from wild-derived and laboratory mice. For the first time, class-wise inter-observer reliability of manual labels used for ground truth are analyzed and reported. Moreover, we developed a new classification method based on ensemble deep learning that provides more generalizability than the current state-of-the-art tool (both pretrained and retrained). Our new classification method is free for scientific use.

## Introduction

The ultrasonic vocalizations (USVs) of house mice (*Mus musculus)* and rats (*Rattus norvegicus)* are surprisingly complex, and they are increasingly being investigated to better understand animal communication [1–3] and as a model for studying the genetic basis of autism and speech disorders [4,5]. Rodents emit USVs in discrete units called *syllables* or *calls* (these terms are metaphors and do not imply that rodents use words, or that their vocalizations function to attract other mice). USV syllables are separated by gaps of silence and they have been classified into several different categories by researchers visually inspecting spectrograms [1–3,6–10] i.e., the squared modulus of the short-time Fourier transforms (STFT) [11], or, less often, by statistical clustering analyses [12–17]. USVs are classified according to their shape and other spectro-temporal features, including the length of each syllable, their frequency, and degree of complexity. Classification provides the basis for subsequent analyses of USVs, such as repertoire size (e.g., Nicolakis et al. [7], Marconi et al. [6]) and sequences or *syntax* (e.g., Heckman et al. [3],Chabout et al. [18]).

Several classifications of USVs have been proposed, which vary from three to 12 different classes, and there is no consensus on how they should be classified. Researchers agree that there is a qualitative distinction between simple versus complex types of USVs (with the latter having frequency-jumps or harmonics), but not with other proposed classes, as their differences are fuzzy. Many proposed classes are quantitative variations within these two major categories (e.g., simple USVs show quantitative differences in length and shape, and complex syllables can have one or more frequency-jumps). A recent study concluded that USVs do not cluster into distinctive types, and instead form a continuum [19]. However, since USVs are mainly classified by human researchers, the crucial question is how do rodents perceive and respond to variations in USVs. Continuous differences in these calls might still be perceived as categorically discrete by rodents, just as we perceive continuous speech as discrete words and variations in wavelengths of light as different colors. Few studies have addressed questions about perception so far, and the evidence suggests that mice differentiate some though not other USV classes (see Outlook below). Moreover, house mice emit different types of USVs depending upon the social contexts and potential receivers [10,18,20–24], and they alter their syllable type usage over the time during courtship and mating [25–27]. Thus, identifying variations in USVs in different contexts is central to studying the functions of these vocalizations (Nicolakis et al. [7], Marconi et al. [6]).

The main technical challenge for USV processing and analyses includes developing better methods for detecting and classifying these vocalizations, since most analyses are still conducted manually by visual inspection of spectrograms, which is extremely time-consuming.

The first step in this signal processing task is USV detection, which is a challenging problem due to the low signal-to-noise ratio (SNR) in most recording conditions. Manually detecting

each USV can take an enormous amount of time, particularly with large datasets. Semi-automatic methods are useful, but they are still time-consuming (e.g., semi-automatic detection using Avisoft SASLab Pro and manual checks requires 1–1.5 hours to detect merely 150–300 USVs [28], and some datasets contain tens of thousands of USVs [6]). The time required to classify USVs takes even longer than detection, and classification is a necessary step to evaluate qualitative differences in vocalizations and to conduct analyses of USV sequences (syntax) (e.g., von Merten et al. [8]). Several software tools have recently become available for automating USV detection, including MUPET [13], MSA [14], DeepSqueak (DSQ) [16], USVSEG [29], Automatic Mouse Ultrasound Detector (A-MUD) [30], Ultravox (Noldus; Wageningen, NL) (commercial), and SONOTRACK (commercial). These tools enhance the efficiency of processing USV data, but they can generate errors for several reasons. Failing to detect actual USVs (the probability is given by the false-negative rate or FNR) can result in missing actual differences in the vocalizations, and erroneous detections (false detection rate or FDR) can lead to failure to detect actual differences and generate false differences. The challenge for any USV detection algorithm is maximizing the true positive rate (TPR) while minimizing the FNR and FDR. Moreover, automatic methods can have systematic biases depending on how they are developed. For example, automated methods developed using only one mouse strain, one sex, or only in one state or context can increase both types of error when applied to other mice or conditions (see S1 Table for the mice and recording conditions used for developing different USV detection tools if applied in other settings). Thus, automated methods can greatly enhance the efficiency of processing USV data, but it is critical that they can be generalized. Results should be treated with caution until the error rates in the detection and classification method are evaluated for particular datasets, or their generalizability is demonstrated.

To our knowledge, five studies have compared the performance of USV detection algorithms: (1) Binder et al. [28] compared MSA and Avisoft for detecting USVs emitted from different strains of mice (C57BL/6, Fmr1-FVB.129, NS-Pten-FVB, and 129). They concluded that Avisoft outperformed MSA for C57BL/6 and NS-Pten-FVB strains, but these two methods performed similarly for strain 129. Thus, there are strain-specific differences between these two detection tools. (2) Another study [31] compared the quantity of USVs detected by Avisoft to those detected by Ultravox (2.0) and reported significant differences in USV detection and weaker than expected overall correlations between the systems under congruent detection parameters. (3) Van Segbroeck et al. [13] compared MUPET and MSA for detecting USVs emitted by B6D2F1 males from MouseTube [3] and found that these methods generated similar call counts and spectro-temporal measures of individual syllables. (4) Coffey et al. [16] compared MUPET, Ultravox, and DSQ for detecting USVs by analyzing the TPR and precision (the ratio of detected true USVs to false positives). For this purpose, they manipulated a recording from MouseTube in two ways to gradually degrade its quality. In the first experiment, increasing levels of Gaussian white noise were added to recordings, and DSQ outperformed MUPET and Ultravox in terms of TPR and precision in all Gaussian noise levels. In the second experiment, real noise was added to recordings, and DSQ again outperformed MUPET in terms of precision and Ultravox in terms of precision and TPR. (5) Zala et al. [30] compared the performance of Avisoft and A-MUD (version 1.0) in identifying USVs of wild-derived *Mus musculus musculus*. They concluded that the latter method is superior in terms of TPR and FDR. Zala et al. [32] have since provided an updated version of A-MUD, which overcomes previous difficulties in identifying faint and short USVs.

Our first aim was to systematically compare the performance of four commonly used USV detection tools, MUPET, DSQ, A-MUD, and USVSEG, and to determine which is the most efficient and requires the least user intervention. We addressed three main questions:

1. How does the performance of different USV detection methods compare to each other? Previous studies indicate that A-MUD outperforms Avisoft, which outperforms MSA; MSA is comparable to MUPET and DSQ outperforms MUPET and Ultravox. To our knowledge, no study has systematically compared the performance of A-MUD and DSQ, or evaluated more than two of these methods together, though Coffey et al. [16] recently, compared DSQ, MUPET, and Ultravox.

2. How does the performance of USV detection methods compare to ground truth (i.e., manual detection by trained researchers)? Evaluation of detection methods rarely includes such a positive control, which is a crucial comparison to obtain absolute versus relative estimates of performance (e.g., see [30]). Binder et al. [28], Binder et al. [31], and Van Segbroeck et al. [13] compared Avisoft and MSA, Ultravox and Avisoft, and MUPET and MSA based on the number of USVs detected by each of the two methods, but no comparisons were made with ground truth. Coffey et al. [16] used only ca. 100 manually detected USVs as ground truth for comparing DSQ, MUPET, and Ultravox.

3. How well do USV detection tools perform when using novel datasets that differ from the original training set (often called, generalization performance, out-of-sample error, or out-of-the-box performance)? To our knowledge, only one study [28] has tested whether USV detection methods generalize to other mouse strains (comparing only Avisoft and MSA), and only one study has compared MSA and MUPET for different recording conditions (males vocalizing in response to female urine, an anesthetized female, and awake female) [13]. Van Segbroeck et al. [13] and Coffey et al. [16] only used recordings from a hybrid strain (B6D2F1), and Zala et al. [30] used wild-derived *Mus musculus*. Consequently, it is unclear how well current detection methods perform whenever applied to new recordings that differ from the data used to develop and evaluate the tool. The problem of generalization is well known in the machine learning community and there are several approaches to improve "transfer learning" [33].

Therefore, we compared the "out-of-the-box" performance of these USV detection tools with each other, and with ground truth, and we assessed their performance with novel datasets. For these comparisons, we used recordings of laboratory mice (*Mus laboratorius*) and wild-derived house mice (*Mus musculus musculus*), and using recordings under different social contexts and recording conditions. The data were obtained from sources not involved in the developmental phase for our tools (see Data and Methods). To evaluate the absolute performance of these models, we applied a new dataset of manually detected USVs as ground truth with a total of 3955 USVs. We minimized adjusting the detection parameters or re-training these tools because such additional user interventions would add more variables and make it impossible to compare their efficiency. One could include re-training before using or testing a detection tool with a novel dataset, but then the data would have to be re-labeled, which defeats the purpose of using an automated tool. To evaluate performance, we compared TPR (i.e., how often USVs are correctly detected) and FDR (how often background noises are mistakenly detected as USVs). Signal detection theory explains the inevitable trade-off between FPs and FNs [34], and therefore, the most effective tool will provide an optimal balance of these types of errors.

We also aimed to develop an improved method for detecting FPs, as a second refinement or data cleaning step to remove noise before classifying USVs or making other analyses. Whenever analyzing recordings of mice, there are always background noises i.e., non-USV sounds generated from recording instruments or movements of the mouse and bedding especially during social interactions. FNs are problematic as they result in a loss of data for

subsequent analyses; however, false positives from detection are more problematic for statistical analyses and training a classification tool. One can set the parameters of detection such that it errs on the negative rather than the positive set, as FPs can be deleted in the refinement step. To remove FPs, MUPET and DSQ include a preliminary detection refinement step using either an unsupervised approach, which groups data based on similarity measures rather than manually labeled USVs (both approaches), or a supervised approach, which requires manually labeled USVs for training a classifier (DSQ and [35]). Our preliminary evaluation found that DSQ outperformed MUPET in the detection refinement step (using the K-means clustering [36]), however, its performance differs depending on the data. Thus, we designed a method better suited to deal with the problems mentioned above and we compared our method with DSQ for detecting FPs, as this is a critical step for accurate USV classification.

Our second aims were to evaluate the state-of-the-art method for automated USVs classification, and to develop a better method, i.e., an out-of-the-box, high-performance, and supervised method that requires minimal human intervention. Automatic classification of USV syllable types can be achieved through unsupervised [12–14,16,17] and supervised [16] classifiers. The advantage of unsupervised classification (often called 'clustering') is that it is considered to be more objective, as it does not require a predefined number of classes or manually labeled observations. Hence, the number of classes is based on the information contained in the dataset rather than the researchers' assessment. These clusters do not always match the classification of USVs by researchers and it is unclear how they are perceived by mice (see Outlook below). In contrast, supervised classification (classification *sensu stricto*) methods require that researchers first classify or assigning labels to USVs for training a classifier (machine learning), which has higher accuracy compared to clustering [37,38]. One needs to use supervised classification for comparing the results between datasets and manual labels. To our knowledge, only a few studies have used supervised methods for classifying mouse USVs (see S1 Text).

Since the generalizability of USV classifiers has never been investigated (unlike methods for classifying bird vocalizations [39]), it is not known how well the current methods can classify USVs for novel datasets. Again, to evaluate a classification method, a systematic evaluation of a new dataset not used for training or testing is needed. We identified four key factors that can reduce the performance and generalizability of USV classifiers:

1. Noise is a potential problem for classification, as for detection, but this issue has not received sufficient consideration. Some methods used only recordings that had low background noise (high SNR data) for developing and testing their models (e.g., [40], [16], and [41]). This approach seems logical but it results in reduced performance when using more typical recordings of mice having a low-SNR [42]. This problem is exacerbated if the model is developed using predefined features extracted from spectrograms (e.g., see [40]), as the extraction of these features from low-SNR signals already introduces high variance.

2. Imprecise USV detection generates subsequent classification errors. As the main output after detection is usually the time and frequency range of USVs, the classification will only include the region of the spectrogram limited to the detected minimum and maximum USV frequency [16,40]. Our investigations, however, revealed that faint portions of USVs are often not included inside this window, leading to significant errors in feature estimation and classification.

3. Neural networks are being increasingly used for USV classification [16,41,43]. Machine learning is an iterative method and it can fail to find the most effective weights for classification, however, because the algorithm takes a path that reduces the error and this can lead to

focusing on specific weights, which may not be very useful. Becoming trapped in a local minimum is a common problem, and it can reduce the generalizability of a classifier [44]. This problem can be overcome by using ensemble machine learning methods [45], a procedure that uses multiple algorithms, and the final output is obtained from combining the outputs of these models. This approach makes it possible to obtain a model with better performance than any of its component models and it allows for more flexible structures, though developing ensembles require additional training time.

4. Limited training and evaluation inflate model performance. The performance of any model is overly-optimistic whenever the same type of data (e.g., same mouse strain or recording context) is used for both model development and evaluation [40,41,43]. Using such a limited training set conceals the model's shortcomings in dealing with different strains or recording conditions, but surprisingly, previous studies have never considered this issue.

Thus, to develop new and improved methods for USV classification, we had the following aims:

1. Apply a CNN Snapshot Ensemble classifier based on the stochastic gradient descent algorithm, which is accurate even with noisy (low-SNR) data.

2. Use the full time-frequency images based on the entire frequency range and reduce the dimensionality (and thereby the computational load and the possibility of overfitting) using Gammatone filters applied to the spectrograms.

3. Compare our new method with pretrained (as an out-of-the-box model) and retrained DeepSqueak (DSQ), which is currently the state-of-the-art classification tool, and evaluate these methods using USVs recorded under different conditions and from different mice strains than the conditions and strains used in the training step.

## Data and methods

### USV data

**Subjects.** The data used in this study was first divided into two meta-sets: we have used one development set (DEV) to develop, train and test the developed detection and classification methods. To test the generalizability of the methods we use an additional evaluation (EV) set. For a direct test, as well as estimating the meta-parameters of the classifier, using stratified 8-fold cross-validation, the DEV dataset was further divided into three subsets including DEV_train, DEV_validation, and DEV_test (Table 1). We report the performance of the proposed classifier in Sections "Selecting the architecture of the classifier", "Evaluating BootSnap for classifying USVs", and "Inference classification" over the DEV_validation and DEV_test datasets. The DEV dataset (Zala et al. [30]) combined two pre-existing datasets: the first dataset was from 11 wild-derived male and 3 female mice (*Mus musculus musculus*) recorded for 10 min in the presence of an unfamiliar female stimulus [24]. In the second data set, 30 wild-derived male mice (*M. musculus musculus*) were recorded for 10 min in the presence of an unfamiliar female on 2 consecutive days, first unprimed and then sexually primed. These were F1 and F2 descendants from wild-caught *M. musculus musculus*, respectively (which for brevity, we refer to as 'wild mice'), whereas laboratory mice are domesticated hybrids of three *Mus* subspecies, and mainly *Mus musculus domesticus*.

The EV dataset consists of two datasets, and a part was obtained from wild mice ('EV_wild') (as in DEV), but under different conditions [6]. The vocalizations were obtained from 22 sexually experienced adult wild-derived (F3) male *M. musculus musculus* [6]. Male vocalizations

**Table 1. Number of instances for each class in the different datasets.**

| Data set | Number of members in each class | | | | | | | | | | | | | |
|---|---|---|---|---|---|---|---|---|---|---|---|---|---|---|
| DEV_train | c | c2 | c3 | c4 | c5 | h | d | up | u | f | us | s | ui | FP |
| | 308 | 241 | 69 | 0 | 0 | 124 | 299 | 4343 | 298 | 1277 | 74 | 291 | 543 | 4849 |
| DEV_validation | 53 | 42 | 12 | 0 | 0 | 21 | 52 | 753 | 52 | 221 | 13 | 51 | 94 | 840 |
| DEV_test | 50 | 39 | 11 | 0 | 0 | 20 | 48 | 695 | 48 | 205 | 12 | 47 | 87 | 776 |
| EV_wild | c | c2 | split | | | Rise | | | | | | | ui | FP |
| | 20 | 224 | 334 | | | 1025 | | | | | | | 110 | 234 |
| EV_lab | 61 | 404 | 739 | | | 819 | | | | | | | 200 | 389 |

were recorded without and also during the presentation of a female urine stimulus over three recording weeks, one time per week and each time for 15 minutes. To evaluate classifier performance, we used three arbitrarily chosen recordings out of these 66 recordings, and manually classified them for this study. The other part of the EV data is taken from the MouseTube dataset used for developing DSQ ('EV_lab') (B6D2F1 mice recorded by Chabout et al. [18]) and two arbitrarily selected recordings were sampled out of these 168 recordings. Although we only used a few recordings to evaluate the methods, these recordings contained a large number of USVs (Table 1). In order to prevent any potential bias in the performance of our method, we selected 4 datasets that differed in their recording methods and other characteristics, such as recording contexts (males with fresh or frozen female urine, males or females with a stimulus female separated by a divider), subjects' previous experience (males without or with sociosexual experience), microphone used (condenser ultrasound microphone Avisoft-Bioacoustics CM16/CMPA and USG Electret Ultrasound Microphone Avisoft Bioacoustics / Knowles FG) and genetic background of mice (wild-derived mice of F1-F3 generation and B6D2F1/J laboratory mice). See S2 Text for more detailed information on all datasets.

**Detection.** For USV detection, we applied A-MUD (version 3.2) using its published default parameters for both the DEV and the EV datasets. Because FPs and syllables are detected during the detection process, we call the detected segments 'elements' rather than 'syllables'. The parameters that affect A-MUD performance are o1_on, o1_off and if oo is enabled, oo_on and oo_off, which are amplitude thresholds in decibel. For this study, we use two A-MUD outputs: the element time slot and the estimated track of the instantaneous frequency over time (frequency track; FT), called 'segment info' (Fig 1). We also compared A-MUD to the three other detection tools, MUPET, DSQ, and USVSEG. To ensure a comparison, where

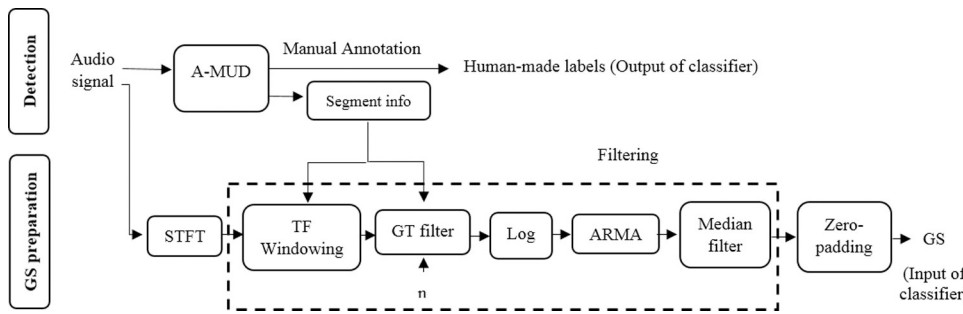

**Fig 1. Block diagram showing the procedure for USV detection and input preparation for the classifier.** $n$ is the Gammatone (GT) filter order. STFT, A-MUD, ARMA, and GS are the abbreviation for short-time Fourier transform, automatic mouse ultrasound detector, autoregressive moving average, and Gammatone spectrograms, respectively. TF in 'TF windowing' is the abbreviation for time-frequency. In this step, we restrict the spectrogram to the time of interest, where the segment is detected, and to the frequency of interest, i.e., 20 kHz to 120 kHz.

A-MUD is certainly not privileged, the parameters of A-MUD were fixed while those of the other approaches were optimized, through trial-and-error, i.e., we used the best parameters, which provide the highest true positive rates for each detection tool, and not the default settings. The parameters used for evaluating the different tools are presented in S2 Table.

Since the detection tools that we compared in this study were developed and evaluated using USVs of wild mice (A-MUD) and laboratory mice (DSQ, USVSEG, and MUPET), we also use USVs from both types of mice for our evaluation (two recordings for wild mice from the DEV and EV_wild + two recordings for the laboratory mice from EV_lab). The DEV_1 (1 sound file from DEV data), EV_wild_1 (sound file 1 from EV_wild data), EV_lab_1 (sound file 1 from EV_lab data), and EV_lab_2 (sound file 2 from EV_lab data) signals consist of 947, 771, 1013, and 1224 USVs, respectively.

**Manual annotation of detections.** After automatically detecting all elements, the DEV dataset was manually classified into 12 classes (Fig 2), depending on the USVs' spectro-temporal features [5–7,9,32,46] (S3 Table). These classes are based on frequency changes [32] (> 5

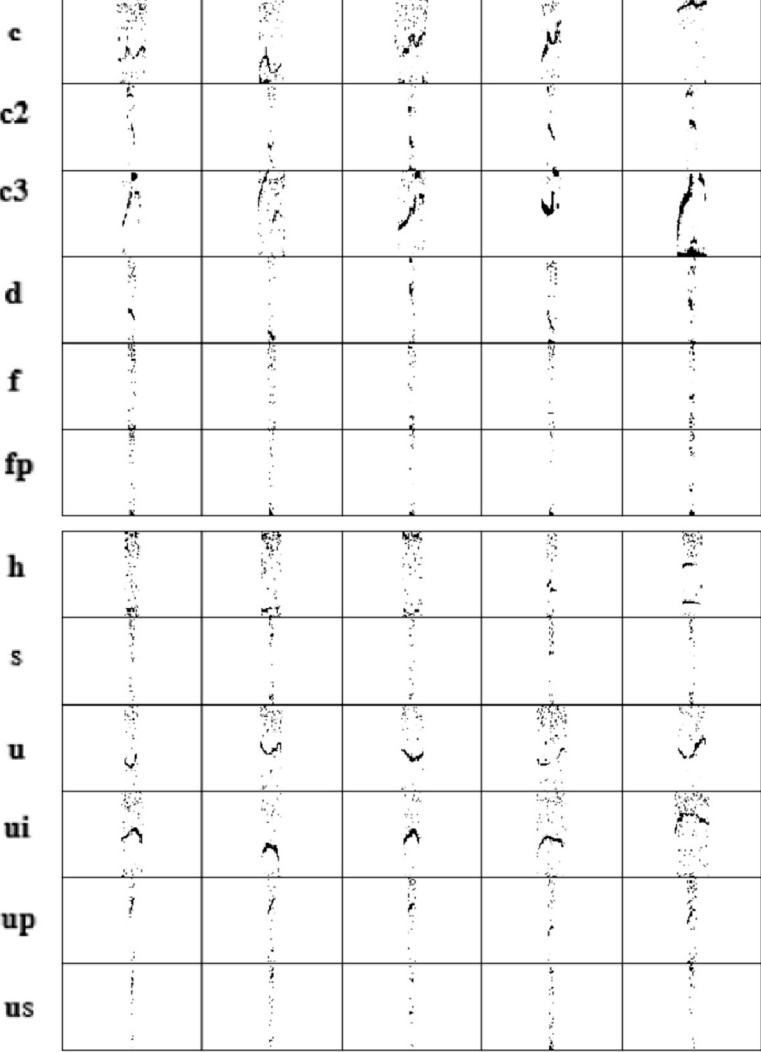

**Fig 2. Gammatone Spectrograms (GSs) of five members of 12 studied classes.** These GSs have the minimum Manhattan distance to other members of 12 USV classes in the development dataset.

kHz increase 'up', > 5 kHz decrease 'd'), on the number of components (corresponding to breaks in the frequency track; 'c2' with 2 and 'c3' with 3 components), on changes of frequency direction ($\geq$ 2 changes 'c') or shape (u-shape, 'u', u-inverted shape, 'ui'), on frequency modulation (< 5kHz, 'f'), on time (5–10 ms, 's', < 5ms, 'us'), and harmonic elements, 'h'. It is worth noting that there are 2 more USV classes, USVs with 4 'c4' and 5 'c5' components. Due to their infrequency, however, they are excluded from the training task (DEV dataset), but they are used for the evaluation step (EV dataset).

When using low-SNR recordings, or recordings with faint or short USVs, certain background noises are sometimes mistakenly detected as USVs. These errors are false positives (FPs), whereas USVs that are missed are false negatives (FNs). As mentioned above, minimizing one of these types of errors increases the other one, due to inevitable tradeoffs in signal detection [47]. FPs are preferable over FNs, as they can be excluded in a follow-up step, and thus we included FP as a target class. The DEV dataset contained 16958 elements including 6465 FPs in total (Table 1).

When comparing our model with DSQ, the EV data (EV_lab and EV_wild) were manually labeled into 6 classes: 'c2', 'split' (pool of 'c3', 'c4', 'c5', and 'h'), 'c', 'ui', 'FP', and 'Rise' (pool of 'up', 'd', 'f', 's', 'us', and 'u'). We created the classes 'split' and 'Rise' because DSQ reported them together with 'c2', 'c', 'ui', and 'FP' as the output classes. The EV dataset consisted of 4500 elements including FP, of which 1947 and 2615 instances belonged to wild mice and laboratory mice, respectively.

**Input images for the classifier.** Handcrafted, predetermined features (such as slope, modulation frequency, number of jumps, etc.) are affected by noise, so the development of a classifier based on these features increases the error of the classification, as discussed in the Introduction. Therefore, we developed an image-based supervised classification built on the STFT of detected elements, followed by a set of filters and a zero-padding method (Fig 1).

After applying the time segmentation obtained from A-MUD, a STFT (NFFT = 750) with a 0.8-overlapped Hamming window is applied to the signals, as shown in Fig 1. The desired information in the frequency interval of 20 kHz to 120 kHz and in the time interval of detected elements is extracted ("TF windowing", Fig 1).

A spectrogram (the squared modulus of the STFT) is often used for the analysis of USVs and machine learning approaches [16,41]. But the problem is that spectrograms lead to high computational demands and, because of redundancy, they pose high risks of model overfitting. Following Van Segbroeck et al. [13], a Gammatone (GT) filter bank [48] was therefore used to reduce the size of the STFT array along the frequency axis from $251 \times 401$ to $64 \times 401$ while simultaneously maintaining the key spectro-temporal features. It can be interpreted as a pooling operator using a re-weighting step, which is motivated by a comparison with filterbanks adopted to human auditory perception [49]. Therefore, we adapted the frequency distribution to make our method applicable to the auditory range of mice.

GT filter bank computations are provided in a MATLAB script by [50]. These computations were converted into the Python language for the present study. For each filter, a central frequency and bandwidth are required. The bandwidth and center frequency equations obtained in MUPET are also employed here (see S2 Text). In MUPET, the midpoint frequency parameter (Eq 2 in S2 Text) used to calculate the central frequencies was chosen as 75 kHz. The midpoint frequency can be interpreted as the frequency region where most information is processed [13]. Because the authors acknowledged that this value may not apply to all mice, we estimated the optimum value by calculating the median frequency (i.e., 63.5 kHz) from the FTs of all detected syllables, omitting FPs (S1 Fig). Then, in a pilot test, we updated this value to 68 kHz to minimize the information loss from USVs. The central frequency was calculated based only on the DEV data. A more detailed explanation of how to determine these two

parameters is given in the S2 Text (the Gammatone filterbank section). To further eliminate the background noise from the images, following MUPET, we calculated the maximum value between the Gammatone-filtered STFT pixels and the floor noise ($10^{-3}$). The logarithm of the output was smoothed using an auto-regression moving-average (ARMA) filter [51] with order 1 (S2 Text). Finally, a median filter [52] was applied to remove stationary noise. The product of the pre-processing is a smoothed, denoised spectrogram with reduced size of 64*401, called Gammatone spectrograms (GSs). Fig 2 shows the GSs of five samples of each 12 studied classes. These samples have the minimum Manhattan distance to other members of each class.

**CNN classifier.** For our study, we used convolutional neural networks (CNNs), a particular form of the deep neural network [53] first introduced by [54] and further developed by [55]. A brief description of how this model works, how we implemented it, and how the DSQ classifier is retrained is provided in the S2 Text.

We have evaluated our classifier for different values of its hyperparameters and architecture to achieve the best performance. These parameters were the number of convolution layers (i.e., 3, 4, and 5), the number of filters in each convolution layer (16, 32, 64, and 96), the kernel size in the first convolution layer (i.e., (3, 3), (5, 5), and (3, 18)), the drop out percentages (i.e., 0.5, 0.6, and 0.7), the size of dense layers (i.e., 32, 64, and 128), and the learning rate (cosine annealing learning rate scheduler [44], fixed learning rate = $10^{-3}$, and decreasing learning rate = ($10^{-3}$ to $5 * 10^{-6}$)).

In this study, we used categorical cross-entropy (CCE) [53,56]. For the reduction of the overfitting [57] $L^2$ regularization [58] is added to CCE. To optimize the loss function, we used the stochastic gradient descent with Nesterov momentum [59] and we initialized the weights of the convolution and FC layers using the He-initialization [60]. To reduce overfitting and to promote the generalizability of the model [61], we performed the augmentation of the training dataset using random shifts of width and height by 10%. We chose the following architecture for the classifier based on the comparison of the model performance on DEV_validation data (S4 Table).

The architecture of our network is shown in Fig 3. In this depiction, e.g., Conv2D (32, 3*18) denotes a 2-dimensional convolution layer with a kernel size of 3*18 and 32 filters. The FC (128) is a fully connected layer with 128 neurons. After two FC layers, a dropout layer with the probability of 0.5 is used. This step reduces the risk of overfitting [62].

**Imbalanced data distribution.** As shown in Table 1, the DEV_train dataset is significantly unbalanced, with 69 occurrences of the 'c3' and 4849 of the 'FP' class, a typical situation in real applications of machine learning. To investigate how this uneven distribution affects

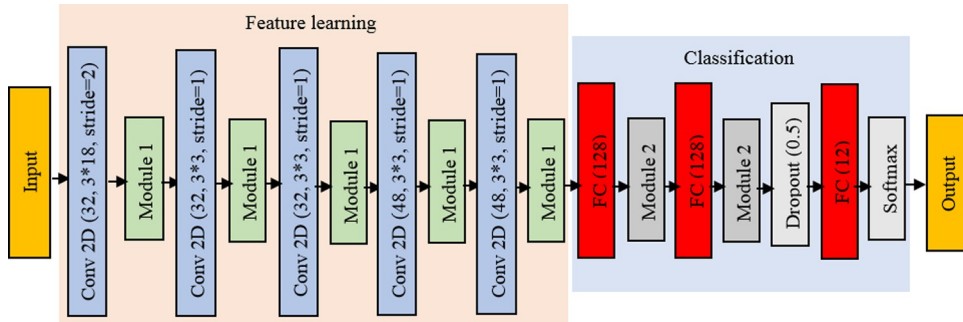

**Fig 3. Classifier architecture.** Module 1 consists of the following layers: Batch normalization + ELU + Maxpooling 2*2. Module 2 consists of the following layers: Batch normalization + ELU. Conv2D (32, 3*18) is a 2-dimensional convolution layer with a kernel size of 3*18 and the number of filters is 32. FC (128) is a fully connected layer with 128 neurons.

the performance of the classifier, in addition to the original DEV_train data, we fit the model with resampled DEV_train using three different approaches.

1. In the first approach, the original input data are bootstrapped *10* times to increase the generalizability and reliability of the classifier [63,64]. Here, we used bootstrap to increase (decrease) the randomness (variance) during the model development. In each bootstrap iteration, samples are drawn from the original dataset with repetition, so some samples may appear more than once or some not at all. Then, we fitted a model for each bootstrapped dataset. The final model performance was evaluated by the average over the 10 models. Bootstrapping reduced the ratio of data imbalance from 76 to 4.

2. In the second scenario, all classes, except the classes 'c3' and 'us', which only have a maximum data number of 69 and 74, are randomly under-sampled to 124 samples.

3. In the last scenario, all classes, except 'FP' and 'up', are over- and under-sampled to the number of samples of the majority class, i.e., 4849. We used the Synthetic Minority Oversampling Technique Edited Nearest Neighbor (SMOTEENN) [65] and the number of neighbors was selected as 3.

To tackle the imbalanced distribution, during the model training we also weighed the loss function inversely proportionally to the number of class members [66] for the original, bootstrapped, and under-sampled data using the following equation:

$$WCCE = -\sum_{i=1}^{C} cw_i \, y_i \, log \, log \, (p_i), \;\; where \; cw_i = \frac{N}{c * n_i} \tag{1}$$

N and $n_i$ are the total number of samples and class members. CCE [53,56] in Eq 4 in the S2 Text was updated to WCCE.

**Model ensemble.** The weights optimized on a particular dataset are not guaranteed to be optimal (or even useful) for another dataset. At the same time, different machine-learning algorithms can lead to different results even for the same dataset. In ensemble methods [45] the final output is taken from combining the outputs of different models and thus reducing the variance of the classifier output. Rather than training a model from scratch for different sets of hyperparameters, we produced 5 trained models during the training of a single model using Snapshot Ensemble with cosine annealing learning rate scheduler [44]. The use of the Snapshot Ensemble does not add complexity to the classifier, whereas it does help to take advantage of ensemble learning without needing to train additional models. The ensembles were trained consecutively, so the final weights of one model are the initial weights of the next. In this approach, the CNN weights are saved at the minimum learning rate of each cycle (S2 Fig), which occurs after every 40 epochs. To determine the best combination of these 5 models, we have cross-validated 4 approaches: 1) using the predictions of the 5th model, 2) using the average prediction from the last 3 models, 3) combining the predictions of the last 3 models by Extreme Gradient Boosting Machines (XGBMs) [67], and 4) combining the predictions of all 5 models using XGBMs. In explaining the third and fourth methods, instead of taking the average of the predictions (used for the second method), the predictions of the last three and five models of the DEV_validation data together with their ground truth are used for training the XGBMs. In this case, the final output of the classifier is the output of XGBMs.

Thus, to develop our classifier, these four ensemble methods were applied for each resampling approach namely under-sampling, over-sampling, and bootstrapping, and for the original data.

**Inter-observer reliability (IOR).** Our ground truth (or 'gold standard') was based on manual classification by researchers, and we used two independent observers to classify USVs

and then to evaluate our ground truth, we evaluated the reliability of our ground using class-wise inter-observer reliability (IOR). The first 100 USVs of 10 sound files were manually classified into 15 USV types by two of the authors, and both have much experience (Marconi et al. [6],Nicolakis et al. [7],Zala et al. [32]). We used five arbitrarily selected sound files from the DEV dataset and all five sound files used for the EV dataset (EV_wild and EV_lab). Both observers were blind to their respective labels and the original labels used for the development or evaluation of our classifier. The USV labels were extracted and exported into *Excel* files. The exported parameters included the start time, end time, and USV type of each vocalization. Then, the labels from both observers were aligned according to the start time of each segment. Thus, vocalizations with the same starting time were compared between the two observers. Segments that were labeled as false positives by the observers but detected by A-MUD as candidate USVs, were included; and segments that were labeled as unclassified ("uc") were excluded from the analyses. Segments classified as the same type by both observers were scored as 'agreement'. Segments that were either detected by only one observer or were classified into a different class were scored as 'disagreement'. Then, we calculated the percentage of correctly classified USVs by both observers, reported as IOR. We calculated the IOR for DEV and EV data for all segments (including FPs), and when including and excluding USVs detected by only one observer and not the other (i.e., labeled as 'missed' USVs). In addition to the original data, we calculated the IOR and F1-score when excluding 's' and 'us' classes, to evaluate how these two classes affected the IOR, and when pooling the original data into 12, 11, 6, 5, 3, and 2 classes, respectively, to compare the IOR and F1-score with the performance of our classifier.

## Results

### Comparing detection algorithms

Fig 4 shows the performance (TPR and FDR) of the four detection tools, MUPET, (pretrained) DSQ, USVSEG, and A-MUD (S1–S4 Data). A-MUD was tested using its default parameters, whereas the others were implemented using the combination of parameters that provided the

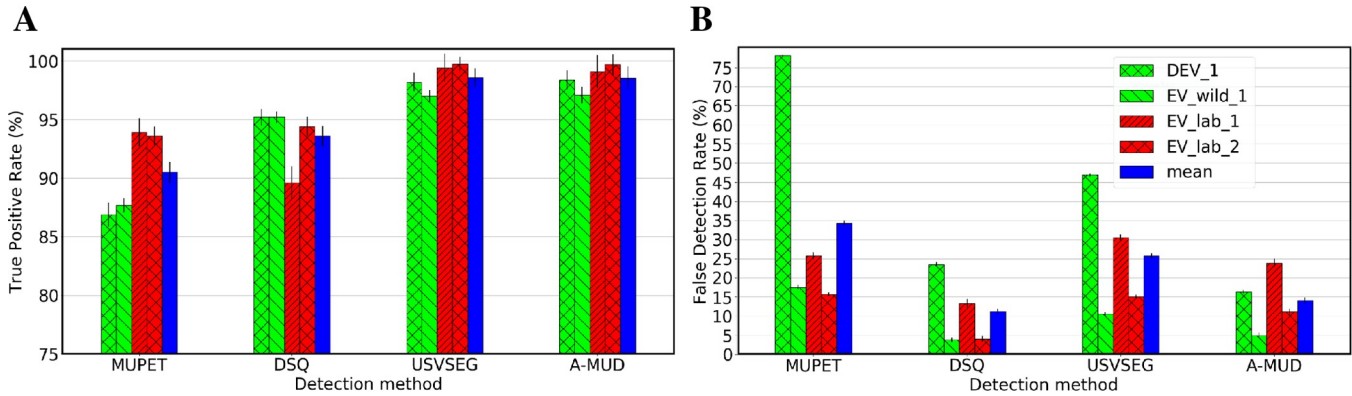

**Fig 4. Best performance of four USV detection methods for four recordings.** (A) The True Positive Rate shows the ratio of the number of USVs correctly detected to the total number of manually detected USVs * 100. (B) The False Detection Rate shows the ratio of the number of unwanted sounds (noise) incorrectly detected as USVs to the total number of detected elements * 100. Error bars represent the estimated variance calculated from the bootstrap resampling method. MUPET was implemented with the noise-reduction parameter set at 2, minimum syllable duration of 5 ms, and a minimum frequency of 30 kHz [13]. DSQ used its detection with the short rat call_network_v2 network with a high "recall" parameter [16]. USVSEG applied its detection with the threshold parameter set at 2.5, the minimum gap between syllables at 5 ms, and the minimum length of USVs at 4 ms [29]. A-MUD was run using its default parameters [30]. The legend shows the four recordings that were compared for each method (i.e., laboratory mice vs wild mice for both DEV (i.e., DEV_1 and EV_wild_1) and EV datasets (i.e., EV_lab_1 and EV_lab_2) and the mean of these four recordings. DEV_1 and EV_lab_1 are examples of low-SNR recordings and EV_lab_2 is an example of high-SNR recording.

best results for the chosen dataset. We also compared the performance of these methods using other parameters (S3 Fig).

A-MUD (with the default parameters) and USVSEG (with the tuned parameters) correctly detected the largest number of USVs (TPRs were all >97%) whereas MUPET had the lowest mean TPR (90%) (Fig 4). A-MUD and USVSEG also provided the best performance when evaluating the detection of USVs from low-SNR recordings (DEV_1 and EV_lab_1, which include USVs from wild-derived and laboratory mice, respectively). We evaluated the performance of USVSEG using recordings of laboratory and wild mice and found that it has a higher TPR for laboratory mice using any of its settings (S3 Fig). This result is likely because this method is primarily parameterized and evaluated based on recordings of laboratory mice. In contrast, A-MUD (with the default parameters) has a high TPR for both types of data, despite that it was parameterized and evaluated using recordings of wild mice only. The presence of faint USVs (in EV_wild_1) had little effect on the TPR for most methods, except for MUPET. The TPR for this method was reduced from 93% to 86% when recordings contained faint USVs. By comparing FDRs, we found that DSQ had the lowest error rates, though it has fewer mean TPR than A-MUD (93.6% vs 98.6%). This shows that users need to be aware of the limitations of using these tools (like DSQ) without re-training and fine-tuning.

Visual inspection of the results indicates that the highest variance of TPR (~ 1.2%) and FDR (~1%) when comparing all the tools occurred in the data EV_lab_1. The TPR of USVSEG reached A-MUD (98.6%), whereas it underperformed A-MUD in terms of FDR (25.7% vs. 13.7%). Also, by examining the output of USVSEG, we found that most of its FPs are fragmented faint USVs, so they do not resemble FPs and, thus, must be manually removed from this group and assigned to the USVs.

Our results also show that A-MUD and USVSEG underestimated the duration of USVs in wild mice and overestimated them in laboratory mice (S3 Text). The slopes (and intercepts) between USV duration estimated by the two tools and observations are not statistically different (permutation test, $p$-values > .05). These results can explain some of the errors in the classification of USVs because overstimulation (underestimation) may cause the inclusion of noise (removal of useful information) in the USV segmentations. Further investigation of this error is beyond the scope of this paper.

## Selecting the architecture of the classifier

To develop our classifier, the detected elements were first manually classified into 12 types of USVs (ground truth). In addition to the original data, three types of resampling approaches were examined (under-sampling, over-sampling, and bootstrapping) to overcome the uneven distribution between USV classes. For each type of resampling, four model ensemble methods were applied to the outputs, which include the predictions of the last Snapshot ensemble ('sn'), the average prediction of the last 3 Snapshot ensemble models ('sn_avg_3'), and a combination of the predictions of the last 3 ('sn_xgb3') and 5 Snapshot ensemble models ('sn_xgb5') by XGBMs. To investigate the effect of snapshot ensemble and bootstrapping approaches on model performance, we considered the classifier trained using a learning rate of $10^{-3}$ (called 'single model' in Fig 5) and original data as the baseline. Fig 5 shows the performance of the models with different combinations of resampling and ensemble methods compared to the baseline model.

The comparison of F1-score obtained from baseline model (68.9±2.3%) and model trained using Snapshot ensemble (based on original data) (70.6±1.4%) shows the superiority of Snapshot ensemble. In addition, bootstrapping data (without using Snapshot ensemble) increased the F1-score by about 6% compared to the baseline model. The bootstrap and under-sampling

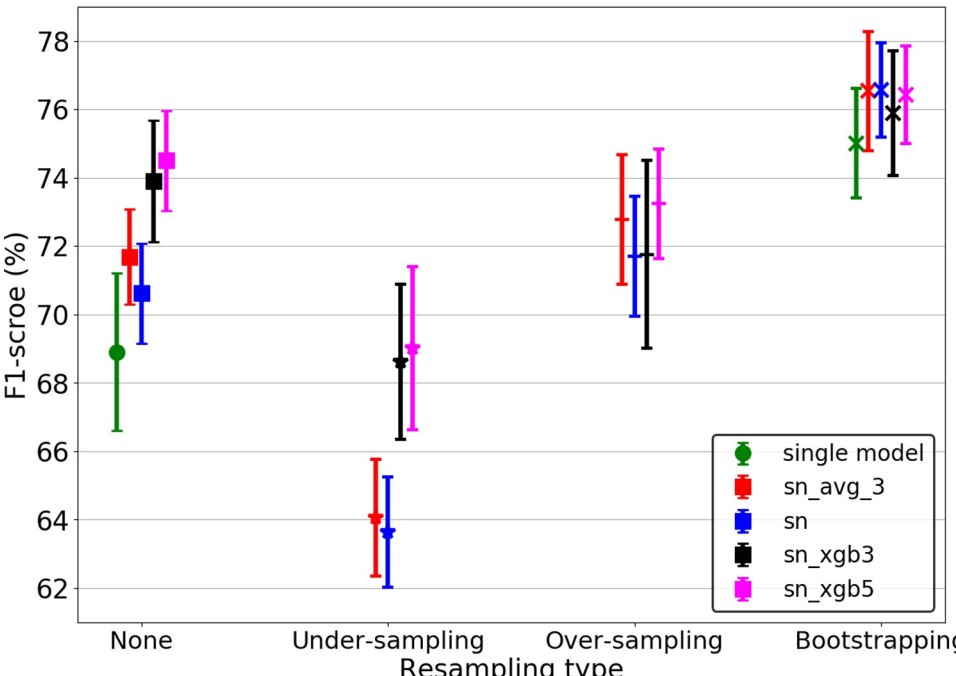

**Fig 5. Performance of classifiers based on four resampling methods for four types of ensemble models.** The single model performance is only displayed in two types of resampling (including 'none' and 'bootstrapping'), to better understand the effect of the 'bootstrapping' approach on the baseline model (which is based on original data and fixed learning rate). Using snapshot ensemble-based method, for each type of resampling (including 'none', 'under-sampling', 'over-sampling', and 'bootstrapping'), four ensemble models have been applied to the outputs last Snapshot ensemble ('sn'), the average prediction of the last 3 Snapshot ensemble models ('sn_avg_3'), and combining the predictions of the last 3 ('sn_xgb3') and 5 Snapshot ensemble models ('sn_xgb5') by XGBMs. The mean ± STD of macro F1-score of test datasets over 8-fold cross-validation are shown.

methods always had the highest and lowest average F1-score, respectively, regardless of the ensemble method. Using the last model obtained from the Snapshot ensemble gave the highest average F1-score (76.6%) with bootstrapping. 'sn_xgb5' outperformed the other ensemble methods for the original data and two other resampling methods (under-sampling and over-sampling). The last model of the Snapshot ensemble also provided the lowest variation in boot-strapped data (STD = 1.4%). The differences between the ensemble methods are not large if used together with bootstrapping.

Neither the under-sampling (F1-scores = 69%) nor the over-sampling (F1-scores = 73.5%) methods improved the performance of the model compared to the best model from the original data (F1-score = 74.5%). While this result is not surprising for the under-sampled case, the performance of the oversampling case shows that the variance is here not a problem for small classes. The poor performance of the model fed by under-sampled data can be attributed to the random discard of samples and thus the deletion of useful information. The over-sampling method may have failed to improve the model performance because the images produced by the SMOTEENN are very similar to the original data (S4 Fig) leading to model overfitting. As a result, the combination of bootstrapped data and the last Snapshot model (hereafter called *BootSnap*) provided the best classifier.

Next, we examined the class-wise performance of the best model for each combination of resampling and ensembling method, including original + 'sn_xgb5', under-sampled + 'sn_xgb5', over-sampled + 'sn_xgb5', bootstrapped + 'sn' (*BootSnap*), and baseline model. As shown in Fig 6, *BootSnap* improved the F1-scores of classes 'c2', 'up', 'ui', 'c3', and 'us' by

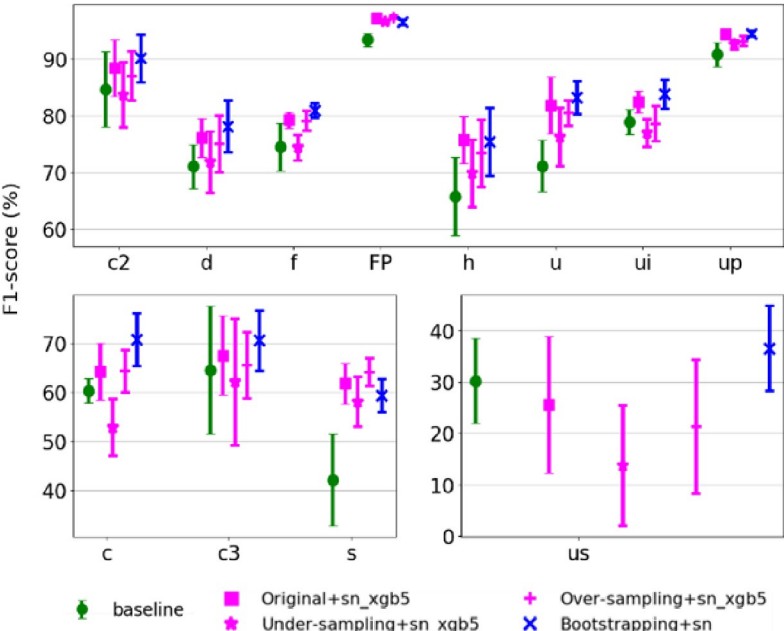

**Fig 6. Performance of baseline model and the best model for each combination of resampling and ensemble methods.** The performance of the baseline model (trained using a learning rate of $10^{-3}$ and without the use of resampling and ensemble methods) and the best model resulting from each ensemble method for different USV classes are shown. The mean ± STD of the class-wise macro F1-scores is based on the 8-fold cross-validation.

around 5% and classes 'd', 'h', 'u', 'c', and 's' by around 10% or more. It also improved the F1-score of classes 'c' and 'c3' by around 5% and class 'us' by around 10%, compared to other combinations of resampling and ensemble methods. The number of classes 'c3' and 'us' in the original data is lower than in other classes, and bootstrapping seems to effectively increase the number of class members used during the model development. For classes, 'c2', 'd', 'f', and 'u', *BootSnap* increased the average macro F1-score by around 2%-3%. The classes 'FP', 'h', 'ui', and 'up' in the original + 'sn_xgb5' and *BootSnap* models have approximately equal average macro F1-score. Using the XGB output ensembling for bootstrapped data and SMOTEENN increased model complexity and did not improve the performance of the classifier.

Somewhat surprisingly, the average macro F1-score of the classes 'h' and 'ui' did not increase by bootstrapping, so it seems that the number of these data points is sufficient for our method. It appears that bootstrapping did not help only for the class 's', but the abundance of class members of 'up' and 'FP' in the original data defused the effect of bootstrapping. The average macro F1-score of *BootSnap* in the class 's' is about 2% less than in the model fed by the original data.

*BootSnap* also reduced the variation in the macro F1-scores for almost all USV classes, and the largest reduction in variation was for classes 'u', 'c3', and 'us'. However, the classes 'us' and 'c3' had the highest macro F1-score STD in all resampling methods; a result that might be due to the very low number of samples in these two classes (99 and 93 members respectively).

## Evaluating *BootSnap* for classifying USVs

To evaluate the performance of *BootSnap* for different types of USVs, we generated a row-wise normalized confusion matrix (or error matrix) [68] (Fig 7). To prepare this matrix, we used

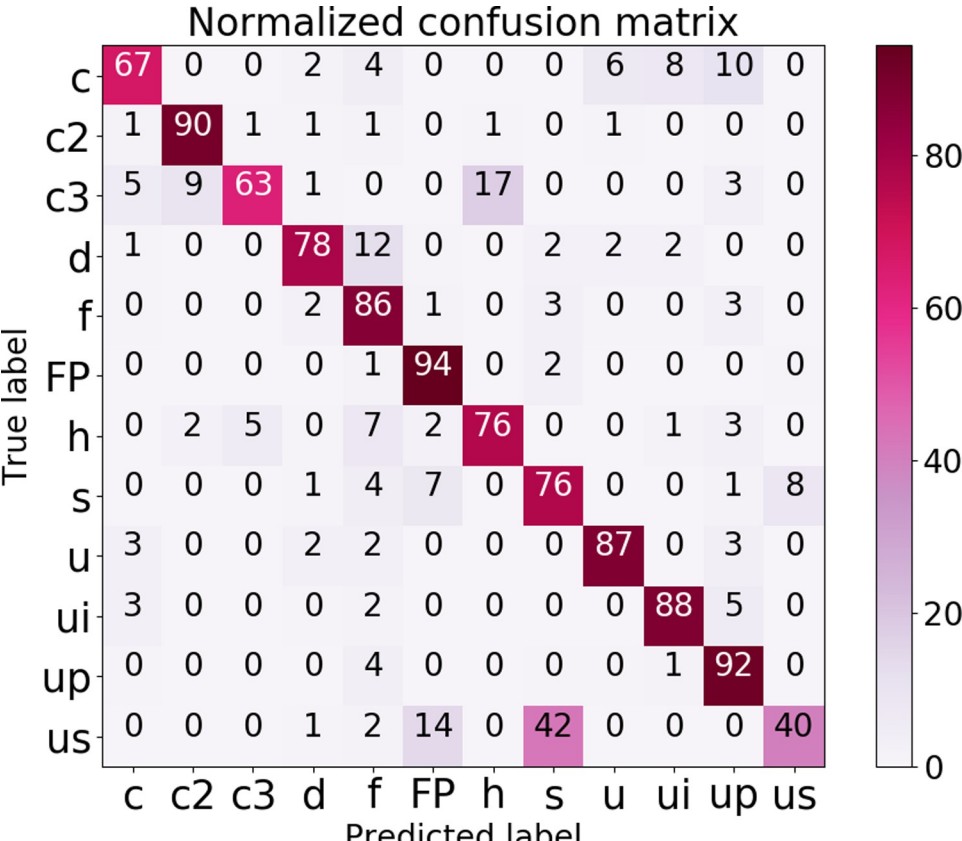

**Fig 7. Confusion matrix of a 12-class classification using *BootSnap*.** The main diagonal represents the recall of each USV class. The other values in each row are FNRs, which indicate the percentage of each class of USVs incorrectly labeled or classified.

the manual annotations and predicted labels from *BootSnap* of the DEV_test dataset (of 8-fold).

This matrix shows that non-USVs ('FP') were classified with the highest recall (94%), which indicates that our model can successfully detect most falsely identified signals, and exclude them from further processing. It also shows that 40% to 92% of different types of USVs were accurately classified. The lowest recall was the 'us' class, and more than 40% of 'us' were mistakenly labeled as class 's' and 14% of the total members were assigned to the class 'FP'. The classification of 's' syllables (76%) was much more accurate than 'us', and the highest FNR value of this class ('s') belongs to the class 'us'. The misclassification of these two classes can be attributed to the use of the USVs length as the only feature used for manual classification, which is not reliable ('us' also shows much lower inter-observer repeatability in manual classification than other classes; S5 Fig). Class 'c3' had the second-lowest recall (63%), and most of its FNs were found with the classes 'h' (17%), 'c2' (9%), and 'c' (5%). These errors were due to the occurrence of harmonic patterns or faint jumps in the class 'c3'. The class 'c' had the third-lowest recall (67%), despite having a high number of members. The problem is that 'c' syllables were often mis-assigned due to their similarity in the spectrograms to 'ui', 'u', and 'up' types, which resulted in the highest FN rates in these three classes. Examination of the misclassified members of the class 'h' indicates that they were often assigned to the class 'f'. The highest portion of FNR (17%) of the class 'c3' is found with the class 'h'. The FNR of the class 'h' is 5%

with class 'c3'. In other words, the members of the class 'c3' are much more likely to be mistaken as the class 'h' than vice versa. It is because harmonic patterns are frequently seen with the second element (out of three elements) in the class 'c3', whereas the opposite rarely occurred in our recordings.

As shown in Fig 2, members of the class 'd' resemble the members of class 'f', which resulted in the class 'd' having the most FNs with the class 'f'. While there is no distinguished pattern of FNs distribution in other classes, it is important to note that FNs of the classes 'c2' and 'c3' mostly occur among themselves. Thus, the performance of the classifier is improved after pooling the 'c2' and 'c3' classes, as we show next.

## Inference classification

Since it is unclear whether and how mice classify USVs, we report the performance of the best classifier (*BootSnap*) based on the different number of classes proposed in previous studies (Table 2). It is important to note that, unlike previous studies, we considered 'FP' as a target class. Since *BootSnap* was trained using 12 classes, we pooled different types of calls in various combinations, especially for the most similar types of syllables, to compare its performance with existing literature treating other numbers of classes. This comparison provides some insights into the classification of types of USVs by researchers.

The number of USV classes studied here ranged between 2 and 12 different types. As expected, classifying all 12 classes provided the lowest F1-score (76.6 ± 1.4%). In the next step, the classes 'us' and 's', which differ only in their duration, were pooled to form a new class, labeled 'short'. By combining these two classes, we expectedly found a significant increase in the F1-score (81.1 ± 1.6%). In addition, by combining these two classes, a significant number of 'us' and 's' types, which were mistakenly assigned as each other (Fig 7), were correctly classified as 'short'. In the next step, the classes 'up', 'd', 'f', 's', 'us', and 'u' were pooled to form the class called 'Rise', and the classes 'c3' and 'h' were included in the class 'split'. Aside from the class 'u', a common feature between classes pooled into 'Rise' was having no changes in their frequency direction. These classes were mostly false positives in the 12-member classification, and thus, after pooling, the F1-score significantly increased to 86.7±1.9%, compared to the 11-class classification.

Then, according to Wang et al. [69], the number of classes was reduced to five. We pooled the classes 'ui', 'c', and 'Rise'. These classes have no jumps in their spectrograms and thus the pooled new class was labeled 'no-jump'. Also, the classes 'h' and 'c3', which were pooled in the previous step into the class 'split', were separated again, but unlike the previous steps, the F1-score decreased (ca. 0.2%). This result might have been due to the separation of classes 'h' and 'c3' causing a large number of members of the latter class to be classified in the former

**Table 2. BootSnap performance in classifying the DEV_test dataset in various combinations of classes.**

| Basis of classifications | # of classes | Different combinations of syllable types | | | | | | | | | | | | Adapted from | F1-score (%) |
|---|---|---|---|---|---|---|---|---|---|---|---|---|---|---|---|
| original | 12 | FP | up | d | f | s | Us | u | ui | c | c2 | c3 | h | [32]* | **76.7±1.4** |
| Pool 's' and 'us' | 11 | FP | up | d | f | Short | | u | ui | c | c2 | c3 | h | [5,46] | **81.1±1.6** |
| - | 6 | FP | Rise | | | | | | ui | c | c2 | split | | [16] | **86.7±1.9** |
| Simple/complex | 5 | FP | no-jump | | | | | | | | c2 | c3 | h | [69] | **86.5±2.2** |
| F- jumps | 3 | FP | no-jump | | | | | | | | jumps and harmonics | | | [10] | **95.4±0.6** |
| FP/USV | 2 | FP | USV | | | | | | | | | | | - | **97.1±0.4** |

*There are more references for 12 classes classification including [46], [5], [7], [6], and [9].

**Table 3. Comparison of pretrained DSQ (out of box model), retrained DSQ, and BootSnap performances.** The performance metric is calculated based on supervised classification of USVs in EV_wild and EV_lab recordings. The values of macro F1-score (which is the average of F1-score over all classes) and class-wise F1-score (F1-score computed for each class) together with their resampling-based variance estimation are presented.

| Scheme | macro F1-score (%) | Class-wise F1-score (%) | | | | | |
|---|---|---|---|---|---|---|---|
| | | c | c2 | split | FP | Rise | ui |
| | *EV_wild* | | | | | | |
| pretrained_DSQ | 41±1 | 0±0 | 44±3 | 56±3 | 50±3 | 82±1 | 12±3 |
| retrained_DSQ | 66±2 | 30±8 | 50±3 | **83±1** | **98±1** | 92±1 | 41±5 |
| *BootSnap* | **67±1.6** | **35±6** | **58±3** | 58±3 | 93±1 | **92±0** | **66±4** |
| | *EV_lab* | | | | | | |
| pretrained_DSQ | 49±1 | 24±3 | 43±3 | 74±1 | 66±2 | 69±2 | 16±4 |
| retrained_DSQ | 40±1 | 8±2 | 48±2 | 53±2 | 54±2 | **71±1** | 6±2 |
| *BootSnap* | **64±1** | **38±4** | **93±1** | **84±1** | **77±1** | 61±2 | **28±3** |

class (Fig 7). In the next step, all the members of the classes 'c2', 'c3', and 'h' were pooled into the class 'jumps and harmonics' and compared with the classes 'FP' and 'no-jump'. As mentioned before, all the FNs of the classes 'c2' and 'c3' were from each other (Fig 7), and as a result, pooling them in one class yielded an F1-score of about 95.4 ± 0.6%. Finally, we classified syllables and 'FP' into two separate classes, and this simple binary classification, which was mostly used in the USV detection step, was able to differentiate USVs from FPs with an F1-score of 97.1 ± 0.4%. These results again show how the performance of *BootSnap* depends upon the type of USV, and that pooling certain classes results in better accuracy. Pooling the USV classes in various combinations provides future researchers with a basis to compare their classifiers producing a different number of target classes with *BootSnap*.

## Comparing *BootSnap* and DSQ: transferability to new datasets

We compared the performance of *BootSnap* to DSQ, which we consider to provide the state-of-the-art classification tool, and we used the EV_wild and EV_lab signals (Table 3 and S5 and S6 Data). As discussed in the Data section, the EV_wild data were obtained from wild-derived house mice (as in DEV), but under different conditions [6] and EV_lab data were from the MouseTube dataset (which is used for developing the original DSQ). To fairly evaluate the performance of the DSQ classifier, we have evaluated both the out-of-the-box (pretrained) and retrained models. In the out-of-the-box model, we have used classifier weights obtained from the original DSQ paper. In the retrained model, we used the classifier weights obtained from training the DSQ classifier using DEV data (S2 Text). Note that for *BootSnap* we have used the weights that were learned with the original training data (DEV_train). So, we did not retrain DSQ and BootSnap based on EV data, because this would then be a new learning approach and not an evaluation of the generalizability of the two approaches. *BootSnap* predictions were pooled into 6 classes, which included 'Rise', 'split', 'ui', 'c2', and 'c' (DSQ reported them as the output classes), and 'FP'. DSQ distinguishes FPs from USVs using a post hoc denoising network [16] before the classification step. For comparison, we considered 'FP' as one of DSQ's final outputs. Since *BootSnap* was developed based on 8 folds, we used the mode of 8 predictions to compare it with the DSQ output. It is also important to note that A-MUD was used to detect USVs in both algorithms to provide a fair basis for comparing the classification step in DSQ and *BootSnap* (this improved the average detection rate of DSQ by 5%).

As expected, all three methods–*BootSnap* and pretrained and retrained DSQ–performed better for the types of mice that were used to train them (wild mice for BootSnap, laboratory mice for the pretrained DSQ, and wild mice for the retrained DSQ, respectively; Table 3). DSQ had F1-scores of 41% (pretrained) and 66% (retrained) for wild mice and 49% (pretrained)

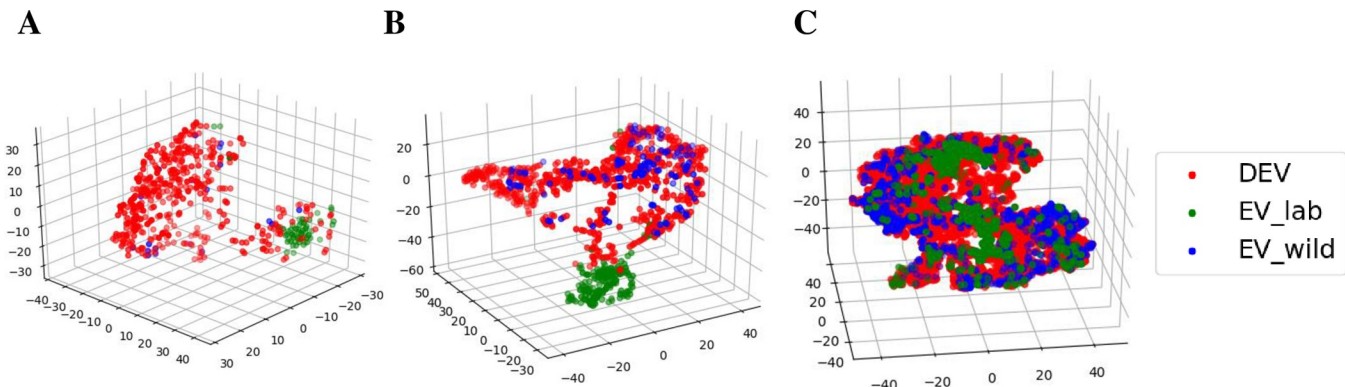

**Fig 8. Scatterplots of USVs from three classes comparing different types of data and mice.** 3-dimensional t-distributed stochastic neighbor embedding (t-SNE) representation of the classes (A) 'c', (B) 'ui', and (C) 'Rise' are shown. Colors indicate the dataset to which USVs belong.

and 40% (retrained) for laboratory mice. *BootSnap* had an F1-score of 67% and 64% for wild and laboratory mice, respectively. Nevertheless, *BootSnap* outperformed pretrained and retrained DSQ for both types of mice overall. In terms of class-wise performance, *BootSnap* performed better than pretrained DSQ in nearly all the classes ('c', 'c2', 'split', 'FP', and 'ui', with higher F1-scores of 32%, 14%, 2%, 43%, and 54% for the EV_wild and higher F1-scores of 14%, 50%, 10%, 11%, and 12% for the EV_lab). The pretrained DSQ outperformed *BootSnap* for the EV_lab for one class, 'rise'. Retrained DSQ outperformed *BootSnap* in the classes 'split' and 'FP' in the EV_wild and the class 'Rise' in EV_lab. Regarding resampling-based variance estimation, the classes with a higher F1-score have less variance, which indicates that the result of that class (e.g., class 'split' in EV_lab) is more consistent.

Once again, an important point for developing and assessing the performance of a classifier is its generalizability, i.e., how well the model works when classifying data not used for the model development. In reviewing the above results, we observed that both DSQ and *BootSnap* had a relatively poor performance in the classification of the classes 'ui' and 'c'. Further examinations showed that the decline in their performance in these classes was due to the significant distance between new data and their training data. This distance is best seen in the three-dimensional t-distributed stochastic neighbor embedding (t-SNE) [70] representation (using the initial dimension of 40, the perplexity of 50, and the number of iterations of 2000) on vectorized GSs from both the DEV and EV datasets shown in Fig 8. The F1-scores of 'ui' and 'c' classes were very low for both *BootSnap* and DSQ for laboratory and wild mice, still, *BootSnap* outperformed DSQ. In the class 'Rise', there was large overlap between the USVs of wild and laboratory mice, which is in contrast with the classes 'ui' and 'c' (Fig 8C). Thus, the performance of both models for this class was much better than for other classes.

As a side remark, not directly linked to the topic of this paper, let us note that considering the data representation in Fig 8, the data from wild mice (DEV and EV_wild) and laboratory mice (EV_lab) could be effectively clustered using the samples from the 'c' and 'ui' classes.

## Inter-observer reliability

When calculating the inter-observer reliability (IOR), excluding 'missed' segments, for the DEV dataset (n = 631 segments from 5 soundfiles) (S7 Data), we found ca. 80% agreement between two independent observers and ca. 84% agreement for the EV dataset (n = 578 segments from 5 soundfiles) (S8 Data), when including all classes (Tables 4 and S5). The removal of the 'missed' segments from all class combinations has a larger effect on IOR in the DEV

**Table 4. Interobserver reliability and resampling-based variance estimation for the subsets of DEV and EV datasets.** IOR values (in percentage) are given for different combinations of classes. Two IOR values are presented for each data and each combination of classes: IOR including 'missed' segments and IOR excluding 'missed' segments.

| Data | Interobserver reliability in various combinations of classes (%) | | | | | | | |
|---|---|---|---|---|---|---|---|---|
| | Original | Excluding 's' and 'us' | 12 classes | 11 classes | 6 classes | 5 classes | 3 classes | 2 classes |
| DEV | 79.5±1.6 | 83.6±1.6 | 79.5±1.6 | 80.6±1.6 | 83.8±1.1 | 89.2±1.1 | 89.2±0.8 | 92.4±0.8 |
| | 85.6±1.4 | 87.4±1.4 | 85.6±1.4 | 86.8±1.2 | 90.2±1.2 | 96±1.2 | 96±0.9 | 99.5±0.4 |
| EV | 84±1.6 | 85.6±1.6 | 88.7±1.4 | 88.9±1.3 | 90.1±1.3 | 93.2±1.1 | 94.6±1.1 | 97.9±0.7 |
| | 85.7±1.4 | 86.4±1.4 | 90.5±1.4 | 90.6±1.2 | 92±1.2 | 95±1.2 | 96.5±0.8 | 99.8±0.5 |

data than in the EV data. This is probably because most of the USVs in the DEV dataset have low-SNR or they have a lower amplitude compared to USVs in the EV dataset since the EV dataset includes the EV_lab files which usually have a high-SNR. So, in the EV data, the probability of error in the detection tool and observer is lower due to the presence of louder USVs.

Excluding the 'us' and 's' USVs increased the IOR to 84% for the DEV data (9% of the segments excluded) and to 86% for the EV data (3.6% of the segments excluded), respectively. A detailed comparison of the manual classification by the two observers (S5 Fig) showed that the USV types 'us', 's', 'up', 'u', 'h', 'c', 'c3', 'c2', and 'ui' in the DEV dataset and 'us', 's', 'up', 'h', 'c4', 'c5', and 'ui' in the EV dataset accounted for the highest disagreement between observers. The disagreement for the type 'us' was likely due to detection error since 'us' USVs have <5 ms duration and be easy to be overlooked by another observer in noisy recordings. If there is a disagreement in the length of USVs (due to faint USVs or background noise) between observers, an "us" might be classified as 's' and 's' USV might be classified as 'd' or 'us'. We observed a low number of 's' and 'us' types when analyzing the EV dataset, especially within the recordings from laboratory mice (9% of 'us' and 's' in the DEV dataset compared to 3.6% in the EV dataset). Additionally, there can be disagreement between the USV types 'up' and 'ui'. This error is likely to occur due to the threshold of 5 kHz to measure the frequency modulation and used to distinguish between 'up' and 'ui'. USVs with upward frequency modulation of >5 kHz ('up') often ends with a slight downward frequency modulation, which can be close to 5 kHz. USVs often have a lower amplitude at the start or the end of the vocalization, and sometimes it can be difficult to measure the exact frequency modulation in a spectrogram. In summary, the main misclassifications are between 1) 'us' and 's', 2) 'c3' and 'h', 3) 'c3', 'c2', and 'c', 4) 'c', 'ui', 'u', and 'up', and 5) 'd' and 'f'. Usually, the fuzzy transition between the types is the main problem in manual classification. Thus, although USV syllables are discrete, they are not all very discrete, especially when the USVs are classified into a large number of classes (e.g., more than 5 according to Table 2). These findings confirm that the main difficulties of *BootSnap* and the manual classification are similar.

In our datasets, errors in manual classification were mainly due to (i) high background noise, (ii) duration or frequency thresholds used to define USV types, (iii) low or high amplitude of USVs (iv), and "noisy" vocalizations with many frequency-jumps emitted by laboratory mice. The disagreement in the manual classification of certain syllable types highlights the importance of finding a biologically relevant number of different USV classes, which can be reliably differentiated with low error rates by different observers.

Similar to the *BootSnap* F1-score, the IOR (Table 4) and F1-score (Table 5) of IOR data improved as we pooled the classes into fewer groups. For example, the IOR improved from 6 to 5 classes classification in the DEV (from 84% to 89%) and EV (from 90% to 93%) datasets. The improved IOR to 89% (DEV) and 94% (EV) after pooling all USVs with or without frequency jumps suggests that this might be a potential classification method, as this is more

**Table 5. F1-score of the DEV_test and subsets of DEV (DEV_IOR) and EV datasets (EV_IOR) for IOR calculation.** F1-score values (in percentage) are given for different combinations of classes. The numbers provided for DEV_test is the same as the numbers in Table 4. They are presented here again for easier comparison. Since we do not have 'missed' segments in the DEV_test data, these segments are removed when calculating the F1 score of DEV_IOR and EV_IOR datasets.

| Setting | F1-score in various combinations of classes (%) | | | | | |
|---|---|---|---|---|---|---|
| | 12 | 11 | 6 | 5 | 3 | 2 |
| DEV_test | 76.7±1.7 | 81.1±1.6 | 86.7±1.9 | 86.5±2.2 | 95.4±0.6 | 97.1±0.4 |
| DEV_IOR | 73.4±4.5 | 77.6±4.7 | 81.8±5.2 | 80.3±5.6 | 91±2 | 99.2±0.4 |
| EV_IOR | 82.8±3.5 | 83.9±2.7 | 89.7±1.9 | 84.2±3.6 | 97±0.6 | 99.6±0.4 |

reliable between observers compared to a classification using ≥12 USV types. Additionally, manual classification showed an agreement of 92% (DEV) and 98% (EV) when distinguishing between USVs and 'FP' segments. The IOR increased to 99.5% (DEV) and 99.8% (EV) when excluding 'missed' segments.

Table 5 shows that in nearly all combinations of classes, the F1-score of DEV_test data (calculated between ground truth and *BootSnap* output) is similar to the F1-score of EV_IOR and higher than DEV_IOR datasets. The F1-score of EV_IOR and DEV_IOR datasets are calculated between two observers' labels. It can be concluded that the value of the F1-score generally increases with the pooling of the classes, and *BootSnap* classifies USVs with approximately equal accuracy as humans.

## Comparing *BootSnap* and DSQ: sensitivity to new classes

One of the main performance factors of a classifier is how well it deals with classes for which it was not trained. The DEV data does not contain samples from two classes, 'c4' and 'c5'. Therefore, to address this issue, we analyzed the performance of pretrained and retrained DSQ and *BootSnap* focusing on these two classes, which were present in EV_wild data.

The results show that *BootSnap* assigned 68% and 32% of the members of these two classes to the classes 'c2' and 'c3', respectively. It is noteworthy that both 'c2' and 'c3' classes represent jump-included USVs, which is also a prominent feature of the classes 'c4' and 'c5'. Pretrained DSQ (retrained DSQ) assigned 3% (0%), 13% (6%), 46% (93%), 3% (0%), and 35% (1%) of the members of the classes 'c4' and 'c5' to the classes 'c', 'c2', 'c3', 'rise', and 'ui', respectively. Although the class 'ui' is relatively similar to the 'c4' and 'c5' classes based on visual inspection (S6 Fig), the difference is that there is no jump in the class 'ui' to which pretrained DSQ incorrectly assigned a significant number of classes 'c4' and 'c5'. Thus, we conclude that *BootSnap* uses a measure of similarity more fitted to USVs than pretrained DSQ, assigning new class samples to the most similar classes in the training data. The retrained DSQ, like BootSnap, assigned mostly all members of the classes 'c4' and 'c5' to jump-included classes ('c2' and 'c3').

## Discussion and conclusions

The most important and novel contributions of our analyses include the following. First, we evaluated the performance and generalizability of four detection methods with each other, and we also assessed their absolute performance using ground truth. Only a few detection tools have been compared in previous studies, and they did not use a ground truth, or if so, they had a very small sample size. For example, the data for our ground truth consisted of 40 times more samples than the pretrained DSQ detector (i.e., 4000 vs 100), and therefore, our results should be much more robust. We used recordings from both wild house mice (*M. musculus musculus*) and laboratory mice, whereas most USV detection tools are designed (or machines are trained) using data from one or a few strains of laboratory mice. We found that A-MUD provided better overall performance compared to other detection methods, and without the

need for any manual parameter tuning or custom training of the networks. Second, we developed *BootSnap*, a new method for USV detection refinement (removing false positives or data cleaning) and classification, and we compared its performance and ability to generalize to novel datasets with DSQ classifier. We found that our new classification method outperformed DSQ–both the pretrained and retrained model–in nearly all aspects, including USVs of both the wild and laboratory mice. Below we address our main results in greater detail.

## Comparing USV detection tools

Our first aim was to compare USV detection methods and evaluate their relative and absolute performance. We used wild mice, as well as laboratory mice, and we also compared recordings that had background noise (DEV_1 and EV_lab_1 signals) and faint (EV_wild_1) elements. We found that A-MUD and USVSEG detected the largest number of actual USVs (TPRs were all $> 97\%$ with A-MUD's *default* parameters and with the adaptive optimal parameters of USVSEG). DSQ and MUPET had the lowest mean TPRs (94% and 90%, respectively), and the pretrained (out-of-the-box) DSQ detector had the lowest FDR. Although DSQ has a lower FDR than other methods, it failed to detect ca. 6% of USVs on average, and to reduce FNR, one would need to train the detection network with labeled data, which would require manually resizing the window of each segment and its label (noise / USV). Although this can be done graphically in DSQ, it ultimately requires much manual intervention (user input).

USVSEG had a somewhat higher TPR for laboratory mice using any of its settings (99%) than wild mice (96%), and this is likely because USVSEG was primarily developed based on recordings of laboratory mice. A-MUD was parameterized using recordings of wild mice, though it still had high TPRs for both types of data, indicating that it is more generalizable than USVSEG. Unlike A-MUD, which was implemented using its default parameters, USVSEG has different performances for different parameter inputs. For example, in USVSEG, the use of the threshold parameter of 2.5, the minimum gap between syllables of 30 ms, and the minimum length of USVs of 4 ms leads to a significant increase in TPR for wild mouse data (above 90%) and a decrease in TPR for laboratory mice data (approximately 62%). While using a gap of 5 ms leads to improved TPR in both data sets. Another point is that the developers of USVSEG have suggested the user change the threshold parameter between 3.5 and 5.5. But we obtained the best performance of USVSEG for the wild mice data when the threshold was set to 2.5. We compared the performance of USVSEG and A-MUD for estimating the duration of USVs. Both methods underestimated the duration of USVs in wild mice and overestimated them in laboratory mice.

We compared how USVSEG and A-MUD detect USVs to better understand how these methods differ. USVSEG detects USVs using the following steps:

1. it calculates spectrograms using the multitaper method, which smooths the spectrogram and reduces background noises;

2. it flattens the spectrogram using cepstral filtering, which is performed by replacing the first three cepstral coefficients to zero and subtracting the median of the spectrogram (flattening eliminates impulse and constant background noises); and

3. it estimates the level of background noise to make a threshold for the resulting spectrogram.

   In contrast, A-MUD (version 3.2) detects USVs using the following steps:

1. it applies an exponential mean to the spectrograms to reduce the noise contribution;

2. it estimates the envelope of the spectrograms using 6–8 cepstral DCT coefficients;

3. it computes the segmentation parameters, which are the amplitudes (m1-m3) and frequencies (f1-f3) of the three highest peaks in the spectrum for each time position; and

4. it searches for a segment based on 4 threshold values.

The reason that A-MUD (version 3.2) and USVSEG outperformed MUPET is presumably that A-MUD uses flattening rather than spectral subtraction for denoising. On the other hand, it seems that USVSEG in some cases leads to the failure of detection of ultrashort USVs, the false detection of two USVs as a single USV, the false segmentation of one USV as two or more USVs.

To summarize, A-MUD provides better overall performance compared to other methods for detecting USVs in audio recordings and without the need for any parameter tuning or custom training of the networks. For these reasons, we utilized A-MUD for our subsequent USV detection.

## Comparing USV classification methods

Our second aim was to develop a new method for USV detection refinement and classification and compare its performance and generalizability with DSQ. To develop the classifier and to overcome the uneven distribution of classes, we examined three types of resampling approaches, under-sampling, over-sampling, and bootstrapping. For each type of resampling, four model ensemble methods were applied to the outputs: the predictions of the last Snapshot ensemble; the average prediction of the last 3 Snapshot ensemble models; and a combination of the predictions of the last 3 and 5 Snapshot ensemble models by XGBMs. We found that the differences between the ensemble methods are not large if used together with bootstrapping. This result can be interpreted in such a way that the ensemble of the models derived from bootstrapped data is already compensating for the uneven distribution statistically. We used bootstrapped data and the last model of snapshot ensembles as the best classifier ('*BootSnap*'). The classifier had the highest errors with classifying ultrashort ('us') USVs mainly due to their similarity with 's' USVs. These USVs do not differ qualitatively, they are not actually different syllable types, as they differ only in length. Another classification error was due to confusing 'c' and 'c3' syllables. The low recall in classifying 'c3' syllable types was likely due to their small number used for training, and also because some members have a harmonic element, much like 'h' types. The similarity in the spectrograms of 'c' to other classes as 'ui', 'u', and 'up' classes lead to errors in the classification of this class. On the other hand, the model classifies classes 'up', 'FP', and 'c2' with a recall higher than 90% and classes 'ui', 'u' and 'f' with a recall of more than 85%. These classes have a relatively larger number of members compared to other classes ('us' and 'c3') and their spectrograms are relatively different from each other. The overall F1-score of the model increased from 76.7% to 81.1% by pooling 's' and 'us' classes, which resulted in a more robust classification.

We compared the performance of *BootSnap* to the pretrained (out-of-the-box) and retrained DSQ classifier, which is currently the state-of-the-art classification tool. DSQ is a user-friendly and straightforward tool for analyzing mouse vocalizations and the user can train it for their data without the need for programming. In this analysis, however, we examined its generalizability and its out-of-the-box usability for novel data, as most users currently use this tool. It uses a 6-member syllable classification that includes 'Rise', 'split', 'ui', 'c2', 'FP', and 'c' types (i.e., a simpler classification approach based on 6 classes, Table 3). USVs from wild mice as well as laboratory mice were used to evaluate the generalizability of these two classifiers. As expected, in the *BootSnap* classifier (and in the retrained DSQ classifier, as well), the

closer the data is to the training domain, the better the overall performance. It has 87±1.9% F1-score for 6-class classification of USVs on DEV_test data (Table 2), but about 65% F1-score for EV datasets. We found that our new classification method outperformed both pretrained and retrained DSQ classifiers in nearly all aspects, including USVs of both the wild and laboratory mice (macro-F1 score of 66% vs 47% and 49%). Again, it is important to emphasize that the performance of retrained DSQ was worse than pretrained DSQ for EV_lab. The main reason for its reduced performance is likely due to the absence of laboratory mice in the DEV data, which would indicate that DSQ is less generalizable than BootSnap. This difference in performance is mainly because the DSQ classifier was developed using an architecture similar to our baseline model fed by high-SNR data, compromising its performance with new low-SNR recordings. In contrast, we used low-SNR data to develop our classifier and aimed to enhance its ability to generalize. We also used the Ensemble learning method, which is based on the Snapshot Ensemble and Bootstrapped input data for training the classifier. In Ensemble learning, base models are combined to prevent the final model from either overfitting or underfitting, making the model more stable and generalizable. So, the novelty of *BootSnap* comes from a clever combination of bootstrapping approach, snapshot ensemble, and Gammatone spectrograms.

*BootSnap* and the retrained DSQ classifier showed better performance than the pretrained (out-of-the-box) DSQ classifier in assigning new class samples to the most similar classes in training data. For example, our results show that the *BootSnap* retrained DSQ classifier assigned all instances with more than 3 jumps (similar to those not found in the training data) to similar classes with less than 3 jumps. However, the pretrained DSQ classifier allocated 30% of these new samples to the class without any jumps. The success of *BootSnap* as well as the retrained DSQ classifiers in assigning new class samples is due to the similarity of the data used for their development and EV_wild data. Our method also detects noise in new data much more accurately (F1-score of 93±1% vs. about 50±3% for EV_wild and 77±1% vs. 66 ±2% for EV_lab), and thus it is more useful for low-SNR data, which is a common challenge for USVs studies–especially studies aiming to record animals under social contexts. Also, *BootSnap* requires less user intervention to classify USVs, as for USVs classification using DSQ the user must first modify the frequency interval of USVs and then apply the classifier on them. But in *BootSnap*, after performing the detection by A-MUD, the whole interval of 20 kHz-120 kHz is used for classification. Another advantage in using *BootSnap* is that it is based on open-source software (Python) and, thus, it is free of charge, whereas DSQ is based on proprietary software (MATLAB), and requires the purchase of required licenses.

While completing the final draft of our present manuscript, a new tool, called 'VocalMat' [71], was published that detects and classifies USVs into 11 categories. The VocalMat classifier is trained on the USVs of mouse pups (5 to 15 days old) of both sexes of several inbred strains, including C57BL6/J, NZO/HlLtJ (New Zealand Obese), 129S1/SvImJ, NOD/ShiLtJ (Non-obese Diabetic NOD), and PWK/PhJ (descendants from a single pair of *Mus musculus musculus*). It was developed using USVs in the frequency range of 45 kHz to 140 kHz. After contrast enhancement and applying several filters, the authors calculated the spectrogram (with the size of 227*227) of 12954 detected elements. Its classifier is the AlexNet model [72], which was pretrained on the ImageNet dataset. Like other studies, this classifier was not compared with other USV tools and the results on its generalizability were not provided. We evaluated the performance of VocalMat on its test data and found that the average class-wise accuracy is 79%, whereas *BootSnap* yielded an average class-wise accuracy of 83% for classifying DEV_test elements into 11 classes. The differences in the performances of these tools could be due to differences in the test data used for evaluation.

### Evaluating ground truth: inter-observer reliability (IOR)

To our knowledge, this is the first time that the class-wise inter-observer reliability (IOR) of the ground truth, used to assess machine performance, has been evaluated. According to the IOR results, the agreement between two observers in the DEV and EV dataset was 80±1.6% and 89±1.4%, respectively. The mentioned values are related to the classification of segments into 12 classes. The USV classification was based both on A-MUD detections and on segments that were missed by A-MUD but manually detected by one or both observers. A closer look at the results reveals that mislabeling members of the classes 'us' as 's', 'ui' as 'up', and 'c' as 'ui' and to a lesser degree as 'up', and vice versa, is very likely. The reason for these errors in manual classification is their sensitivity to the arbitrary threshold (based on duration or modulation frequency) used in their definitions. In addition, the mislabeling can also occur in class 'h', as this class may include faint harmonic elements. Hence, part of the classification error of automated classification can be attributed to the error in the manual labeling of segments. However, any of these classes can be pooled to improve classification (from 80±1.6% using a 12-class classification to 84± 1.1% using 6-classes or to 92±0.8% using 2-class classification, see DEV dataset in Table 4), and such pooling also improved the F1-score of *BootSnap* (F1-score changed from 77% of 12-class classification to 87% of 6-class and 97% of 2-class classification, Table 2). Thus, pooling some types of USVs together improves the accuracy of *BootSnap*, which is expected since some types are very similar to each other. Consequently, *BootSnap* can be expected to perform better when classifying fewer types of syllables and that *BootSnap* can classify USVs with an accuracy similar to the results obtained from human inter-observer reliability. It is no surprise that the particular USVs that *BootSnap* does not classify well are the same ones that humans fail to show consistency. This result suggests that these types of USVs could just be human inventions, though it is certainly still possible that mice might treat them differently.

### Outlook on USV classification

Our USV classification method is supervised, as with other models, and if users want to retrain the algorithm using their own recordings of mice, then manually labeled data must be provided. And despite the outperformance of *BootSnap* over DSQ, *BootSnap* still has difficulties with classifying new data containing complex USVs (with no jumps), u-inverted, and 1-jump USVs. Considering that our best model is based on the bootstrap technique, the computation time increases as the number of bootstrap iterations increases. By default, 10 repetitions are used for *BootSnap*, which means that *BootSnap* calculations will be 10 times slower than similar models. Because manual labeling of data is a difficult and time-consuming task, it is important to be able to apply a model trained on a single data source to other data as well. So, to further improve the generalizability of a classifier, in addition to implementing the bootstrap technique, we will increase the number of samples in the future by using more recordings. We expect that this approach will increase the predictive power of our classifier.

In summary, our ultimate goal was to develop an automated algorithm that provides an out-of-the-box method whose performance is as good or better than a human observer (manual classification). The human F1-score in EV data was higher than the F1-score of the out-of-the-box *BootSnap* model (89.7±1.9% vs 67%, respectively). Although *BootSnap* does not yet achieve our original aim, this first version provides an advance, as it outperforms other methods, including the state-of-the-art model (DSQ; 47% pretrained and 49% retrained). This leaves room for future research.

We emphasize that USVs have been classified by human researchers based on visual inspection of spectrograms or statistical clustering models, and very little is known about whether or

how mice can discriminate most of the various types of USVs that have been proposed. Mice can distinguish frequencies that differ by only 3% [73], and they can be trained to discriminate between simple versus complex USVs, and among certain variations in shape and frequency [74]. They can be trained to discriminate among USVs depending upon their spectro-temporal similarity, and they discriminate complex calls and up-shapes, but not u-shaped calls [75]. Mice fail to discriminate between synthetic sounds with different shapes, i.e., 'up'- vs. down-shapes [76]; however, the shapes of these synthesized sounds were very different from mouse USVs, and may have lacked characteristics that mice use for discrimination. More studies are needed to describe USVs produced in different contexts, and also determine whether mice can discriminate among different types of USVs. Such perception studies should include recordings with normal ranges of variation of syllable types within and between each category (i.e., mice should be better able to discriminate between- versus within-syllable type variation). Until such studies are conducted, the various types of USVs that have been proposed would be more accurately labeled as *USV variants* or *putative call types*.

## Supporting information

**S1 Table. Types of rodents and recording contexts used in different studies.**
(XLSX)

**S2 Table. The evaluated parameters for different USVs detection tools.** Min-f and max-f in MUPET stand for min-frequency and max-frequency. Min-l in USVSEG, MUPET, and A-MUD stands for min-length.
(XLSX)

**S3 Table. Definition of classes used in the labeling.** Note that the number of members differs before and after down-sampling. $F_e$ is the end frequency, $F_s$ is the start frequency, $F_{max}$ is the maximum frequency, and $F_{min}$ is the minimum frequency. The number of members of each class corresponds to the DEV dataset.
(XLSX)

**S4 Table. Performance of the classifier on DEV_validation data using various hyperparameters.**
(XLSX)

**S5 Table. The number of samples of each class of the observer 1 in DEV and EV subsets for IOR calculation.** In the DEV sub-dataset (n = 5 soundfiles), there are very few samples, i.e., 2 from the classes 'c' and 'c4', 4 from the class 'c3', 5 from the class 'u' and 9 from the class 'h', thus the results of these classes are not very reliable. We found similar results in the EV sub-dataset (n = 5 sound files) where there are very few samples from the classes 'us', 'd', 'c', and 'c5'.
(XLSX)

**S1 Text. USV classification literature review.**
(DOCX)

**S2 Text. Supplementary information on data and method.**
(DOCX)

**S3 Text. Comparing the estimated USV duration by USVSEG (using the optimal parameters) and A-MUD with the observed USV duration.**
(DOCX)

**S1 Fig. (A) Distribution of USVs Frequency Track (FT) values, extracted by A-MUD.** The FT values are related to all detected syllables, omitting false positives. (B) The frequency response of 32 Gammatone filters (we have used 64 filters, but for simplicity 32 filters are plotted here) at the frequency range of 20 kHz to 120 kHz. (C) Two examples of the USVs spectrogram before (top row) and after applying the Gammatone filter and post-processing step (bottom row). This image shows that by applying the preprocessing steps on the spectrogram, the important information of the USVs is not lost, even though the size of the images is reduced from $251 \times 401$ to $64 \times 401$.
(TIF)

**S2 Fig. Schedule scheme used for the learning rate.** Using this scheme of the learning rate, the final weights of the model at every 40 epochs are the initial weights of the model in the next epoch. In this approach, the CNN weights are saved at the minimum learning rate of each cycle, i.e., at every 40 epochs.
(TIF)

**S3 Fig. (A) true positive rate (TPR) and (B) false detection rate (FDR) of detection tools.** In the main text, we compared the performance of 4 USV detection tools (USVSEG, A-MUD, DSQ, and MUPET) in a setting (i.e., input parameters) of which the selected parameters lead to their best average performance for the four-given data (DEV_1, EV_wild_1, EV_lab_1, and EV_lab_2). Here, we compared the performance of these methods using all the combinations used for their parameters (S2 Table). If we want to compare the best performance of each detection tool with the best performance of others, A-MUD and with a slight difference, USV-SEG are in the first and second places, followed by DSQ and MUPET.
(TIF)

**S4 Fig. Samples produced by Synthetic Minority Oversampling Technique Edited Nearest Neighbor (SMOTEENN).** In the model design section, we used various approaches to deal with the problem of the imbalanced datasets, including using original, down-sampled, bootstrapped, and over-sampled data. The following figure presents the over-sampled data by SMOTEENN presented. The first column from the left is the original instance and the next columns are the resampled samples. The first, second, third, and last rows are from the classes 'c', 'c3', 'c2', and 'h', respectively. The images produced by the SMOTEENN are very similar to the original data, so, compared to the original data, this method did not help to improve the classifier performance.
(TIF)

**S5 Fig. Agreement between two observers for two subsets of (A) model development (DEV) and (B) evaluation (EV) data. 'missed' segments are elements that are manually detected by only one observer.** Both figures show high disagreement between the observers for both data in the 'us' and 'h' classes. In more detail, the amount of reliability in the DEV data in 'c3' and 'u' classes is very low. Differently, in the EV data, the reliability is less than other classes in 'c4' and 'c5' classes.
(TIF)

**S6 Fig. Samples of USVs from the classes 'c4' and 'c5', USVs with 4 and 5 jumps, respectively.** As mentioned in the results section (in the main text), the performance of a model is important when dealing with a new class. Because there was no sample of the 'c4' and 'c5' classes in the DEV data, we compared the output of the BootSnap and DSQ methods when the two classes were in the EV data. The following figure shows examples of members of these two classes in EV_wild data. *BootSnap* assigned 68% and 32% of the total members of these two

classes to the 2 and 3-jump included USVs, respectively. DSQ assigned the members of the classes 'c4' and 'c5' mostly to the 2 and 3-jump included USVs and 'ui'. Although the class 'ui' might be relatively similar to the 'c4' and 'c5' classes based on visual inspection, there is no jump in this class.
(TIF)

**S1 Data. Data of 4 studied detection methods (USVSEG, MUPET, A-MUD, and DSQ) on DEV_1 recording.**
(XLSX)

**S2 Data. Data of 4 studied detection methods (USVSEG, MUPET, A-MUD, and DSQ) on EV_wild_1 recording.**
(XLSX)

**S3 Data. Data of 4 studied detection methods (USVSEG, MUPET, A-MUD, and DSQ) on EV_lab_1 recording.**
(XLSX)

**S4 Data. Data of 4 studied detection methods (USVSEG, MUPET, A-MUD, and DSQ) on EV_lab_2 recording.**
(XLSX)

**S5 Data. Data of EV_wild segments classification by BootSnap and pretrained and retrained DSQ compared to manual labels.**
(XLSX)

**S6 Data. Data of EV_lab segments classification by BootSnap and pretrained and retrained DSQ compared to manual labels.**
(XLSX)

**S7 Data. Data of interobserver reliability for the subsets of DEV datasets.**
(XLSX)

**S8 Data. Data of interobserver reliability for the subsets of EV datasets.**
(XLSX)

## Acknowledgments

We would like to thank Anton Noll for making A-MUD outputs available.

## Author Contributions

**Conceptualization:** Reyhaneh Abbasi, Peter Balazs, Dustin J. Penn.

**Data curation:** Reyhaneh Abbasi, Maria Adelaide Marconi, Doris Nicolakis, Sarah M. Zala, Dustin J. Penn.

**Formal analysis:** Reyhaneh Abbasi.

**Funding acquisition:** Peter Balazs, Sarah M. Zala, Dustin J. Penn.

**Investigation:** Maria Adelaide Marconi, Doris Nicolakis, Sarah M. Zala.

**Methodology:** Reyhaneh Abbasi, Peter Balazs.

**Project administration:** Peter Balazs, Sarah M. Zala, Dustin J. Penn.

**Resources:** Reyhaneh Abbasi, Peter Balazs, Maria Adelaide Marconi, Doris Nicolakis, Sarah M. Zala, Dustin J. Penn.

**Software:** Reyhaneh Abbasi.

**Supervision:** Peter Balazs, Sarah M. Zala, Dustin J. Penn.

**Validation:** Reyhaneh Abbasi, Peter Balazs, Maria Adelaide Marconi, Doris Nicolakis.

**Visualization:** Reyhaneh Abbasi.

**Writing – original draft:** Reyhaneh Abbasi, Peter Balazs, Maria Adelaide Marconi, Doris Nicolakis, Sarah M. Zala, Dustin J. Penn.

**Writing – review & editing:** Reyhaneh Abbasi, Peter Balazs, Maria Adelaide Marconi, Doris Nicolakis, Sarah M. Zala, Dustin J. Penn.

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
