## [Decision Letter · Decision Letter 0]

7 Aug 2021

Dear Mrs. Abbasi,

Thank you very much for submitting your manuscript "Capturing the songs of mice with an improved detection and classification method for ultrasonic vocalizations (BootSnap)" for consideration at PLOS Computational Biology.

As with all papers reviewed by the journal, your manuscript was reviewed by members of the editorial board and by several independent reviewers. In light of the reviews (below this email), we would like to invite the resubmission of a significantly-revised version that takes into account the reviewers' comments.

The reviewers broadly agree on the strengths of the paper but have diverging views on whether the paper presents enough novelty or rigour for publication. The most critical review (R2) raises some significant issues, most of which I concur with. Hence I will give an editorial decision of "major revisions". We do not guarantee that a revised article would be published, but there is scope for the authors to address the identified issues.

We cannot make any decision about publication until we have seen the revised manuscript and your response to the reviewers' comments. Your revised manuscript is also likely to be sent to reviewers for further evaluation.

Sincerely,

Dan Stowell

Associate Editor

PLOS Computational Biology

Natalia Komarova

Deputy Editor

PLOS Computational Biology

The reviewers broadly agree on the strengths of the paper but have diverging views on whether the paper presents enough novelty or rigour for publication. The most critical review (R2) raises some significant issues, most of which I concur with. Hence I will give an editorial decision of "major revisions". We do not guarantee that a revised article would be published, but there is scope for the authors to address the identified issues.

Reviewer's Responses to Questions

**Comments to the Authors:**

Reviewer #1: Abbasi et al. provide an extensive manuscript detailing the comparison of out-of-sample performance of several publicly available algorithms for detection and classification of rodent USVs, tested here in specifically mice recordings. The authors address an important gap in the literature, namely a systematic evaluation of generalization of algorithms that have been trained with restricted datasets (either in SNR, elicitation, age/sex/species). This evaluation can contribute to the most effective uptake of automatic USV detection and classification methods that are robust against variation between recording settings. The authors focus on supervised ML approaches, reporting macro F1 scores for overall comparison and detailed micro F1 scores across different ranges of class-numbers. Importantly, they test their models on gold-standard human annotation, for a completely new dataset, and also provide inter-rater reliability between two human experts on this dataset, as a practical upper limit to classification accuracy.

The core of the manuscript is well developed and does not raise major concerns. Methods are described in detail and most metrics use resampling/crossvalidation to arrive at an estimate of variance. The MS could benefit from some style/language editing, and a careful inspection of the in-text references to figures and tables that are sometimes off (e.g. line 1328)

I have several suggestions for small improvements:

- Figure 4, Table 5,6,7 are (mostly) missing variance estimates. Can the authors provide (resampled) variance estimates to support inferential statistical comparisons between groups?

- Figure 5/table 3 can be improved. it seems the AMUD slope + AMUD intercept SE are off by a misplaced decimal separator in the EV_lab subtables? The significance estimates of the slopes are not particularly interesting. Rather, a permutation test assessing the significance of the difference in slopes (and intercepts) in the USV length estimates might be more informative. This particular results feels slightly disconnected from the main story and might be delegated to the supplement, unless it can speak to the us vs. s misclassification discussion later on, but that involves an AMUD vs DSQ comparison if I am not mistaken.

- it is unclear which parameter options have been considered in the hyperparameter optimization of the CNN models. Surely, the authors have used different layer setups, learning rates, dropout percentages. Please report the space of the grid search performed over the hyperparameters, and potentially the DEV validation accuracy attained with some/all combinations.

Reviewer #2: Synopsis

--------

Many researchers are interested in the automated quantification of mouse vocal behavior (ultrasonic vocalizations or USVs), which is useful for the study of models of various diseases and the neural underpinnings of social behaviors, among other topics. In the typical analysis approach in which single "syllables" or "calls" are segmented and then described quantitatively, there are two chief technical challenges which are both active research areas. First, the researcher must automatically and reliably detecting USVs from a stream of audio. Second and more open-endedly, the researcher must produce meaningful and useful quantifications of the resulting USVs. Substantial progress or improved understanding of either subtask would be a valuable contribution to the field.

The authors present two primary contributions: an empirical comparison of our four previously published USV detection methods and experiments exploring the supervised classification of manually-defined syllable types, which includes a proposed method called "BootSnap."

Unfortunately, I do not believe either contribution provides substantial progress or methodological insight. Therefore I do not recommend this manuscript for publication.

Major Comments

--------------

In Supplementary Τable 2, describing the detection parameters for each detection algorithm, only the four pre-packaged networks are described for DeepSqueak. From what I can tell, the intended use case of DeepSqueak is for the user to train a detection network using their own species- and laboratory-specific data. Simply using the pre-packaged networks without any fine-tuning is not the intended use case of DeepSqueak and will likely present an overly pessimistic picture of its performance as it is used in practice. I consider this a major weakness of the manuscript.

Also in Supplementary Τable 2, there are only two sets of parameters tested for MUPET and three for USVSEG. This is a very small number compared to, say, a standard grid search, which may result in a pessimistic picture of these methods' performance. In the abstract, the authors write "[a]ll four methods had comparable rates of detection failure, though A-MUD outperformed the others in terms of true positive rates for recordings with low or high signal-to-noise ratios." This seems to be fairly misleading given that A-MUD clearly has the highest false positive rate among all test sets excluding "DEV_1" (Figure S3). It is possible that the other models' true positive rates would improve to the level of A-MUD's if their false positive rates were allowed to increase. Trying more sets of parameters for each model, perhaps enough to sketch an ROC curve, would be useful in this comparison. As it stands, I think the evidence for A-MUD's superiority is fairly weak.

On a related note, I noticed many of the same authors on the references of the development and evaluation datasets and the A-MUD methods paper. Is it possible that the data used to develop the original A-MUD and its successive versions shares characteristics with the datasets used in this paper, for example distinct microphone or cage characteristics or the genetic backgrounds of mice? If so, this would also present a concern in evaluating the methods. If not, a couple sentences sentences clarifying this would be warranted.

I am unsure what to take away from the results in Figure 5 regarding the duration of detected syllables. The trendlines of A-MUD and USVSEG look very similar by eye and the R^2 differences are fairly minor and heavily influenced by outliers. I see that the differences are significant if each syllable is treated as independent (Table 3), but the result that A-MUD performs better with wild mice and USVSEG with laboratory mice is anecdotal given only two combined groups of datasets are compared.

The abstract states: "We also did a systematic comparison of existing classification algorithms, where we found the need to develop a new method for automating the classification of USVs using supervised classification, bootstrapping on Gammatone Spectrograms, and Convolutional Neural Networks algorithms with Snapshot ensemble learning (BootSnap)." Far from a systematic comparison of existing classification algorithms, a single CNN architecture is tested with a single preprocessing pipeline with all combinations of four resampling approaches and four ensembling methods. The combination of methods tested -- SMOTEENN, bootstrapping, XGBM, Snapshot ensembles -- are not very well motivated given that they add considerable complexity and there is no "standard" CNN model that is tested as a baseline. There are perhaps a few standard CNN models that could serve as a baseline, for example a single model trained with class-weighted cross entropy loss on the original training set. It is clear from Figure 6 that the bootstrapping method increases average F1 scores by a modest 2-3%, but the effect of snapshot learning is never determined because it is present in every model. Other useful comparisons would include predictions using standard spectrograms or various acoustic features as input instead of Gammatone spectrograms. I did not find the techniques in this section novel or the results particularly notable.

In the comparison to DeepSqueak's supervised classification, it is not clearly stated whether the classification network's parameters are trained on the training set or kept at the default values. If the parameters were not trained, then I do not believe this is a fair comparison to DeepSqueak. If they were trained, I am surprised by its poor performance (for example, the F1 score of 0 on the 'c' syllable).

I appreciate the inclusion of inter-rate reliability data in the manuscript.

Minor Comments

--------------

The final paragraph provides an interesting and useful discussion of the perceptual validity of syllable categories. There are also similar uncertainties that can be raised on a purely signal-level basis that could diminish a researcher's interest in a categorical description of syllables (Coffey et al. 2017, Figure S6; Sainburg, Thielk, & Gentner 2020; Goffinet et al. 2021). Discrete or continuous, one would hope that a syllable's description is related to biology in some way -- for example, predictive of social context, sex, age, genotype, or individual. Results of this sort could be more convincing than predictions of experimenter-defined labels that are known to be only loosely biologically relevant.

line 211: "Develop the first classifier based on the CNNs algorithm."

Gradient descent is the algorithm used to train a CNN.

Figure 9: what is being represented by t-SNE? Vectorized spectrograms? Gammatone filtered spectrograms?

Sections 2.2.1 and 2.2.1 are all standard techniques and could be put in a supplement.

Line 443: "These P-values assess whether the coefficients are significantly different than zero"

Should be different than 0 for b0 and different from 1 for b1.

Line 664: "These results again show how the performance of BootSnap depends upon the type of USV, and that pooling certain classes results in better accuracy."

Is this effect driven primarily by the number of classes? For example, if Gaussian noise were clustered by k-means with 12 classes and then the labels of different classes were combined to form smaller numbers of classes, I would expect the F1 scores of classifiers to increase.

Figure 3 typo: "Modul" -> "Module"

Line 209: "... we aimed at the following principles: (1) Develop the first classifier based on the CNNs algorithm..."

Both Ivanenko et al. (2020, Plos CB) and Steinfath et al. (2021, bioRxiv) present CNN classifiers for mouse vocalizations. For birdsong, there is also TweetyNet (Cohen et al. 2020, bioRxiv), which may be adaptable to USVs.

Reviewer #3: The authors systematically compare the performance of four methods to detect rodent ultrasonic vocalizations (USV), while also providing evidence of the success of their own USV automation method, BootSnap. The manuscript is well written, providing a thorough review of the literature in a clear manner. The abstract and introduction clearly state the relevance and importance of the research aims, while also providing details that are helpful for a reader who may implement these methods in the future (e.g. processing time in hours). The author’s third aim (“How well do USV detection tools generalize and perform when using data that differs from the training set (by generalization or out-of-sample error)?”) sets this study apart from the other studies that have compared the performance of USV detection methods in the past (e.g. Coffey et al. 2019), while also expanding on past studies by evaluating the 4 published methods (MUPET, DSQ, A-MUD, and USVSEG) in one robust study (aim #1). I applaud the authors for focusing the manuscript so that the practical applications are clear, particularly addressing the major issues that future researchers will likely face when applying new recordings to these detection methods. However, the introduction is lengthy and could be shortened (especially after stating the initial three aims) by moving text to either the methods or discussion sections, depending on the word limit.

The methodology does target the main questions appropriately, although I cannot comment on the machine learning methods (2.1.4, 2.2.1-2.2.3), as this is not my expertise.

The results are presented clearly and logically and appear to be justified by the data. The figures and tables are all appropriate to answer the main aims and the legends are thorough. However, as with the other sections, the results are lengthy and the manuscript could potentially benefit from moving sections to the supplementary material. Ultimately, it is clear that BootSnap outperforms DSQ but it is still not entirely clear how transferable it will be to new data sets, particularly in a wild setting, which is difficult to predict.

The conclusions spelled out in the discussion do respond to each of the main aims and provide even more context by reviewing relevant literature. My only complaint is that the importance of and context for studying USVs is not spelled out, which would improve the manuscript. Overall, the manuscript is long but ambitious, providing many important details to answer each of their aims.

**Have the authors made all data and (if applicable) computational code underlying the findings in their manuscript fully available?**

Reviewer #1: **No: **authors have not yet published data + code (but state they will upon acceptance)

Reviewer #2: None

Reviewer #3: Yes

PLOS authors have the option to publish the peer review history of their article (what does this mean?). If published, this will include your full peer review and any attached files.

Reviewer #1: **Yes: **Marijn van Wingerden

Reviewer #2: No

Reviewer #3: No
---

## [Decision Letter · Decision Letter 1]

30 Nov 2021

Dear Mrs. Abbasi,

Thank you very much for submitting your manuscript "Capturing the songs of mice with an improved detection and classification method for ultrasonic vocalizations (BootSnap)" for consideration at PLOS Computational Biology.

As with all papers reviewed by the journal, your manuscript was reviewed by members of the editorial board and by several independent reviewers. In light of the reviews (below this email), we would like to invite the resubmission of a significantly-revised version that takes into account the reviewers' comments.

My thanks for your sincere work on the manuscript in response to the reviews. Reviewer 2 has continuing concerns about the novelty of the contribution and the fairness of the comparisons. I find these concerns persuasive, and so I recommend major revisions: to be accepted, a resubmission would need to address these substantially.

We cannot make any decision about publication until we have seen the revised manuscript and your response to the reviewers' comments. Your revised manuscript is also likely to be sent to reviewers for further evaluation.

Sincerely,

Dan Stowell

Associate Editor

PLOS Computational Biology

Natalia Komarova

Deputy Editor

PLOS Computational Biology

My thanks to the authors for their sincere work on the manuscript in response to the reviews. Reviewer 2 has continuing concerns about the novelty of the contribution and the fairness of the comparisons. I find these concerns persuasive, and so I recommend major revisions: to be accepted, a resubmission would need to address these substantially.

Reviewer's Responses to Questions

**Comments to the Authors:**

Reviewer #1: I am satisfied with how the authors addressed the issues that I raised, and with the additional analyses. In relation to the points made by R2, I would like to see a good-faith effort by the authors to compare trained models for DSQ and BS on the evaluation datasets. The comparison of the difference between non-retrained and retrained models will shed light on the out-of-the-box and ultimate potential of both methods on these particular out-of-sample validation sets. In order to make the most of the test sets, I would recommend to test the retrained models under crossvalidation.

Reviewer #2: 1) I agree that it would be valuable and convenient to have automated tools that work "out of the box," with no experiment-specific model training. The clarification of this desideratum in the abstract and introduction is helpful. The results in Figure 4 suggest that USVSEG and A-MUD have high enough true positive rates (>96% for three unseen datasets) to be used out of the box for syllable detection for many applications, provided that additional steps are taken to remove false positives. However, I do not believe the results for syllable classification are strong enough to merit the use of pre-trained classifiers as the authors propose. If I'm not mistaken, the results in Table 3 indicate how well the pre-trained classifier generalizes to new datasets. The 6-class macro F1 scores here of about 66%, with individual F1 scores as low as 28%, are well below the estimated 90% achieved by manual labels (Table 5).

2) Even in the revised manuscript, it is not stated clearly that DeepSqueak is used without fine-tuning the network, in a way that was not evaluated by its developers in the original DeepSqueak paper. It is fair to point out, however, that of seven papers using DeepSqueak for syllable detection only two retrained the DeepSqueak detection network, demonstrating that the method is not always used in the best ways in practice. However, of the papers cited that don't fine-tune the network parameters, some perform additional checks were performed to verify the performance (e.g. Hernandez-Lallement et al., 2020). For what it's worth, I have used DeepSqueak before and found it fairly straightforward and user-friendly. Additionally, the authors propose a process in which a network is repeatedly retrained with new user input to cut down on the total amount of user input needed. Even if the authors' intent is to compare out-of-the-box methods, it would be useful as a point of comparison to include a fine-tuned DeepSqueak detection network.

3) Due to the authors' response to my question about training the DeepSqueak classifier, I now realize that the quantitative comparison in Table 3 is not a fair comparison. As stated in the DeepSqueak paper under "Supervised neural network-based classification," the default DeepSqueak classification network was trained using labels produced by an unsupervised k-means clustering algorithm. These clusters were then simply assigned names post-hoc by visually inspecting the corresponding spectrograms. The authors' network, on the other hand, was trained using manual labels produced by the same experts that produced the labels for model evaluation. Therefore, the pre-trained DeepSqueak classifier had no access to manual labels, let alone the labels of the same expert labelers who produced the evaluation labels, unlike the BootSnap classifier. I consider this a major flaw of the manuscript.

4) The addition of the baseline CNN model is an improvement, showing that the combination of snapshot ensembles to the baseline model improves performance. It is unclear whether bootstrapping alone, however, explains the performance of the best classifiers.

5) As the authors note (lines 212-221), neural networks are increasingly being used to classify mouse USVs. I do not believe simply training a new neural network model using low-SNR data (lines 228-229) adds substantial novelty to the methods (DeepSqueak detection networks were trained using artificially low-SNR data). Neither is using Gammatone filters (MUPET uses Gammatone filters). And adding ensemble methods, although possibly useful, is not particularly novel either.

6) The addition of more hyperparameter choices for the detection algorithms is an improvement and did, indeed, improve detection performance.

Summary: I think the comparison of USV detection algorithms presented in Figure 3 is a useful demonstration of the abilities of USVSEG and A-MUD compared to MUPET, although I have issues with how DeepSqueak is presented as explained above. The neural network classification method is not particularly novel and the comparison against the pre-trained DeepSqueak network is not a fair one. Additionally, the results are not strong enough to justify the "out-of-the-box" classification approach the authors propose. The inclusion of inter-observer reliability data was useful in drawing this conclusion and is good practice as well. Overall, despite several improvements of this manuscript over the first submission, I maintain that it is not suitable for publication.

**Have the authors made all data and (if applicable) computational code underlying the findings in their manuscript fully available?**

Reviewer #1: **No: **it seems the code repository is not yet shared

Reviewer #2: **No: **Authors will make code available upon acceptance

PLOS authors have the option to publish the peer review history of their article (what does this mean?). If published, this will include your full peer review and any attached files.

Reviewer #1: **Yes: **Marijn van Wingerden

Reviewer #2: No
---

## [Editor Report · Decision Letter 2]

22 Mar 2022

Dear Mrs. Abbasi,

We are pleased to inform you that your manuscript 'Capturing the songs of mice with an improved detection and classification method for ultrasonic vocalizations (BootSnap)' has been provisionally accepted for publication in PLOS Computational Biology.

Best regards,

Dan Stowell

Associate Editor

PLOS Computational Biology

Natalia Komarova

Deputy Editor

PLOS Computational Biology

---

## [Editor Report · Acceptance letter]

19 Apr 2022

PCOMPBIOL-D-21-01088R2 

Capturing the songs of mice with an improved detection and classification method for ultrasonic vocalizations (BootSnap)

Dear Dr Abbasi,

I am pleased to inform you that your manuscript has been formally accepted for publication in PLOS Computational Biology. Your manuscript is now with our production department and you will be notified of the publication date in due course.

With kind regards,

Olena Szabo
